# Cannabis, Endocannabinoids and Brain Development: From Embryogenesis to Adolescence

**DOI:** 10.3390/cells13221875

**Published:** 2024-11-13

**Authors:** Ricardo J. Rodrigues, Joana M. Marques, Attila Köfalvi

**Affiliations:** 1CNC-UC-Center for Neuroscience and Cell Biology, University of Coimbra, 3004-504 Coimbra, Portugal; joanammarques@gmail.com; 2CIBB-Center for Innovative Biomedicine and Biotechnology, University of Coimbra, 3004-504 Coimbra, Portugal

**Keywords:** CB_1_ receptor, CB_2_ receptor, TRPV_1_ receptor, GPR55, brain development, neurogenesis, neuronal migration, axon pathfinding, synaptogenesis, cannabis

## Abstract

The endocannabinoid signalling system (ECS) plays a critical role from the very beginning of embryogenesis. Accordingly, the ECS is engaged early on in nervous system development, starting from neurulation, supported by the identification of ECS components—both receptors and enzymes controlling endocannabinoid metabolism—at these early stages. In particular, regarding the brain, the ECS is involved in the tightly regulated sequence of events that comprise brain development, from neurogenesis to neuronal migration, morphological guidance for neuronal connectivity, and synaptic circuitry refinement. The importance of this broad role of the ECS across various brain development processes is further underscored by the growing understanding of the consequences of cannabis exposure at different developmental stages. Despite the considerable knowledge we have on the role of the ECS in brain development, significant gaps in our understanding remain, particularly regarding the long-term impact and underlying mechanisms of cannabis exposure at different developmental stages. This review provides an overview of the current state of knowledge on the role of the ECS throughout brain development, from embryogenesis to adulthood, and discusses the impact of cannabis exposure, especially during adolescence—a critical period of circuitry maturation and refinement coinciding with an increased risk of cannabis use.

## 1. Introduction to Cannabis and the Endocannabinoid System

### 1.1. Cannabis Use: A Worldwide Perspective

Cannabis was long considered the most popular illicit drug, but recent legalisation and decriminalisation efforts have been changing its status. Recreational cannabis use is now legalised or at least tolerated in 10 countries across five continents. Certain regions of the United States and Australia have also adopted legalisation, while 27 additional countries have decriminalised its use. Medical marijuana is permitted in more than 50 countries globally. Cultural shifts and expanded legalisation in recent years have fuelled a surge in cannabis consumption across diverse demographics, with an estimated 200 million people worldwide now using it regularly [1]. The rescheduling of cannabis by the US Drug Enforcement Administration (DEA) from the most restrictive Schedule I designation to the less restrictive Schedule III [2], alongside increasing global legalisation and decriminalisation, has been linked to a modest but significant rise in use, particularly among adolescents and young adults [3,4].

While some studies report significant increases primarily among young adults [5,6], discrepancies in data may reflect the brief time frame since legalisation, which is insufficient to fully assess long-term societal trends after an initial surge of interest from novelty seekers. Nevertheless, clear evidence points to an increase in both recreational and medical cannabis use among pregnant women during early pregnancy, the perinatal period, and lactation [7,8]. Reported prevalence rates of cannabis use during pregnancy vary widely due to differences in population and methodology, with stigma and underreporting further obscuring the true extent.

This review underscores the critical role of the endocannabinoid system (ECS) in foetal and adolescent neurodevelopment, highlighting the potential risks that cannabinoid exposure poses to cognitive and emotional health during sensitive developmental windows. By bridging current knowledge gaps, this review aims to inform researchers, healthcare professionals, and policymakers about the pressing need to understand the impact of cannabis use on the developing brain, especially amid its increasing prevalence among pregnant and adolescent individuals.

### 1.2. The Discovery of the Endocannabinoid System

The endocannabinoid system (ECS) was discovered during efforts to understand how marijuana produces its recreational and medicinal effects on the human body. Cannabis research gained momentum during the era of the hippies when Gaoni and Mechoulam elucidated the chemical structure of the two principal phytocannabinoids, cannabidiol (CBD) and Δ^9^-tetrahydrocannabinol (Δ^9^-THC) [9] (Figure 1). Over the last 30 years, it became evident that Δ^9^-THC is primarily responsible for the recreational (psychotomimetic) effects of marijuana despite the plant producing over 120 additional phytocannabinoids [10]. Δ^9^-THC interacts with various receptors in the human body, though only the canonical cannabinoid receptors, CB_1_ and CB_2_ (CB_1_Rs and CB_2_Rs) (Figure 1), are part of the sensu stricto ECS. Through CB_1_R activation, Δ^9^-THC elicits effects such as hypolocomotion, catalepsy, hypothermia, and analgesia, collectively known as the tetrad model in drug-naïve subjects [9,10]. Importantly, other hemp variants with low levels of Δ^9^-THC acid are neither illicit nor psychotomimetic. In contrast, CBD, the other principal phytocannabinoid, is not only devoid of psychoactivity but also antagonises the effects of Δ^9^-THC in most biological assays [10,11,12]. The term “endocannabinoid” was coined 30 years ago to distinguish cannabinoids produced by the body from synthetic and phytocannabinoids [13]. The most studied endocannabinoid messengers are the lipophilic N-arachidonoyl-ethanolamine (anandamide or AEA) and 2-arachidonoyl glycerol (2-AG) (Figure 1). Both can activate CB_1_R and CB_2_R, which are located in various cell types and at different subcellular regions in neurons, at the cell surface and intracellularly [14,15,16,17,18,19,20,21]. It is worth noting that 2-AG is considered the principal eCB agonist in the brain, with signalling-competent levels that exceed those of anandamide by at least an order of magnitude [14,21].

In addition to CB_1_R and CB_2_R, several other receptors, both on the cell surface and intracellularly, are influenced by cannabinoids. One such receptor is the G protein-coupled receptor 55 (GPR55), an L-α-lysophosphatidyl-inositol (LPI) receptor (Figure 1) that shares a modest (13–14%) sequence homology with CB_1_R and CB_2_R [10,27]. Another key receptor is TRPV_1_, a polymodal sensor that responds to heat and toxins, including chilli pepper’s capsaicin, protons, and voltage, and functions as a Na^+^/Ca^2+^ channel (Figure 1). Both GPR55 and TRPV_1_ interact with eCBs, synthetic cannabinoids, and phytocannabinoids, making them important targets for medical cannabis formulations and Epidiolex, an antiepileptic medication based on CBD [24]. Some argue that GPR55 and TRPV_1_ should be considered bona fide endocannabinoid receptors, even though they also play key roles in other signalling systems (Figure 1).

### 1.3. Cannabinoid Receptors

CB_1_R was the first identified and remains the most significant cannabinoid receptor, with high expression levels in the brain. Initially, CB_1_R was detected in cholecystokinin (CCK)^+^ GABAergic interneurons in the rodent and human brain [28,29], but later studies identified CB_1_R in various other cell types, including VGLUT1^+^ glutamatergic cells, monoaminergic neurons, certain cholinergic neurons, astrocytes, and microglia [10,14,15,30] (Figure 1). In contrast, CB_2_R was long regarded as “the peripheral cannabinoid receptor”, absent from the healthy brain. However, in the past two decades, its presence and function in neurons have been increasingly accepted [19,20,31,32,33] (Figure 1). Although no study has systematically compared CB_1_R and CB_2_R densities across brain cell types, it is widely accepted that hippocampal and neocortical GABAergic interneurons are among the highest CB_1_R density in the brain, while pyramidal neurons have much lower CB_1_R levels. Probably even lower levels are found in other neuron types and astrocytes. Marginal CB_1_R expression is expected in microglia, oligodendrocytes, oligodendrocyte precursor cells, and adult neural stem cells [30,34].

Both CB_1_R and CB_2_R engage with various intracellular signalling pathways, depending on the cellular context. These receptors predominantly couple with inhibitory G_i/o_ proteins. Activation of CB_1_Rs and CB_2_Rs typically inhibits adenylyl cyclase, and depending on the cell types, it can stimulate pathways such as focal adhesion kinase (FAK), extracellular signal-regulated kinase 1/2 (ERK1/2), p38 mitogen-activated protein kinase (MAPK), c-Jun N-terminal kinase (JNK), c-Src kinase (Src), neutral sphingomyelinase (N-SMase), ceramide synthesis, and phosphoinositide 3-kinase (PI3K)/Akt. These pathways are crucial for cytoskeletal reorganisation, proliferation, migration, and cell survival or apoptosis. Additionally, via the G_i/o_ βγ subunit, CB_1_R and CB_2_R can inhibit voltage-gated Ca^2+^ channels and activate inwardly rectifying K^+^ channels, leading to membrane hyperpolarisation in neurons [10,35,36,37] (Figure 1).

CB_1_R and CB_2_R often form heteromeric complexes with other G protein-coupled receptors (GPCRs), resulting in novel functional entities with unique responses to cannabinoids, which play an essential role in brain development [38]. One example is the CB_1_R-CB_2_R heteromer, where the unilateral activation of either receptor stimulates Akt/PKB phosphorylation, ERK1/2 activation, and neurite outgrowth in transfected neurons and globus pallidus slices [23]. CB_1_R can also form heteromers with receptor tyrosine kinases (RTKs) that are critical for growth and development [39]. One example involves the transactivation of the TrkB receptor of brain-derived neurotrophic factor (BDNF) via Src kinase in CCK^+^ GABAergic interneurons of the developing hippocampus and cortex [40] (see below) (Figure 1).

GPR55 was first discovered in humans in 1999 and soon emerged as a potential third metabotropic endocannabinoid receptor [37,41]. GPR55 activation by Δ^9^-THC, AEA, 2-AG, and other endogenous LPI-like ligands triggers coupling with G_α12_, G_α13_, or G_αq/11_, leading to increased intracellular calcium levels or the activation of β-arrestin, PKCβII, ERK, p38 MAPK, PLC, RhoA and ROCK [10,38,41,42,43] (Figure 1). GPR55’s involvement in regulating cell proliferation, growth, migration, metabolism, and survival has garnered significant interest in cancer research [44,45]. These functions suggest that GPR55 could play a role in brain development, although GPR55 knockout (KO) mice show no macroscopic brain abnormalities [46] (see below).

Among the many members of the “transient receptor potential” (TRP) superfamily of ligand-gated ion channels, TRPV_1_ serves as an ionotropic receptor for several cannabinoid ligands. It is activated by AEA, 2-AG, and their close relatives, including N-arachidonoyl dopamine (NADA) and N-oleoyl dopamine (OLDA)—both belonging to the so-called endovanilloid class—as well as botanical substances such as capsaicin, CBD, and resiniferatoxin [10,47,48]. The TRPV_1_ channel is composed of four subunits that form a central pore, which is permeable to Na^+^ and Ca^2+^ (Figure 1). These six-transmembrane-domain subunits are prone to alternative splicing, often resulting in functionally distinct TRPV_1_ receptors [49,50]. Notably, the TRPV_1_b splice variant is strongly expressed in the human foetal brain, suggesting a role in development [51]. Both the presynaptic density and functional role of TRPV_1_ receptors decline in the first weeks of postnatal life [52], further supporting the hypothesis that TRPV_1_ may play a developmental role.

### 1.4. Endocannabinoids

#### 1.4.1. 2-Arachidonoyl Glycerol (2-AG)

In the brain, 2-AG synthesis primarily involves the action of diacylglycerol lipases α and β (DAGLα and DAGLβ) [53,54]. Consistent with the mechanism of retrograde 2-AG signalling, DAGLα is postsynaptic and colocalises with dendritic markers in both rodent and human brains [55,56]. Typically, 2-AG synthesis is triggered by postsynaptic Ca^2+^ entry and activation of G_q/11_-coupled metabotropic receptors such as the mGluR5, which in turn activates phospholipase Cβ1 (PLCβ1), releasing sn-2-arachidonoyl-DAG, the precursor of 2-AG [53,57,58]. Postsynaptic Ca^2+^ elevation also activates DAGLα, cleaving 2-AG from its precursor. Although this describes “on-demand” synthesis, evidence supports the existence of a basal synaptic pool of pre-synthesised 2-AG, stored in adiposomes, that is readily releasable [53,59] (Figure 1).

In brain homogenates, monoacylglycerol lipases (MAGL 1 and 2) are responsible for 85% of 2-AG degradation, with the remaining 15% hydrolytic activity attributed to α/β hydrolase domain 6 (ABHD6; 4%) and α/β hydrolase domain 12 (ABHD12; 9%) [54,58,60] (Figure 1).

#### 1.4.2. Anandamide (AEA)

Anandamide is synthesised through several pathways, most notably from N-acylphosphatidylethanolamine by NAPE-specific phospholipase D (NAPE-PLD) [13], as well as by other enzymes such as protein tyrosine phosphatase non-receptor type 22, and through a multi-step process involving α/β hydrolase domain 4 (ABHD4) and glycerophosphodiesterase GDE1 [10,59,61] (Figure 1).

While several enzymes can degrade anandamide, the bulk of its metabolism is carried out by fatty acid aminohydrolase-1 (FAAH-1), which hydrolyses anandamide into arachidonic acid and ethanolamine [10,62]. Humans also possess FAAH-2, an enzyme functionally similar to FAAH-1 but with only 20% sequence similarity [63]. Additional enzymes such as COX-2 and cytochrome P450 are involved in anandamide degradation [10,62] (Figure 1). 

## 2. Cannabinoid Receptors and Brain Development

The involvement of ECS in embryogenesis starts from the very beginning, controlling gametogenesis, fertilisation, oviductal transport, blastocyst development and implantation, entailing a fine-tuned regulation of CB_1_R and CB_2_R activity tightly controlled mainly by precise AEA levels at this early stage [64,65,66,67,68,69]. A precise tone of ECS was also shown to be required in normal trophoblast stem cell proliferation and differentiation [70,71,72,73], being involved in placentation via CB_1_R [73]. In the inner cell mass, embryonic stem cells express both CB_1_R and CB_2_R [74,75,76], significantly up-regulated with differentiation and associated with cell survival [74,77,78]. Mouse embryonic stem cells also express TRPV_1_R, but its role, if any, remains to be defined [76]. This increased expression of CB_1_R and CB_2_R, along with differentiation, is reflected in their involvement in cell lineage commitment and the development of the germinal layers [74,79]. Accordingly, it was shown in chick embryos that the exposure to Δ^9^-THC analogue, O-2545, at gastrulation impaired the formation of brain, heart, somite, and spinal cord primordia [80], corroborated by recent studies in zebrafish also showing that the exposure to Δ^9^-THC and/or CBD during gastrulation induces several later developmental defects including in nervous system development [81,82,83,84]. Such exposure induced alterations in neural plate formation and patterning, indicating a most likely involvement of ECS in the neurulation process [80]. Interestingly, an interaction between cannabinoid signalling and morphogenetic factors [85,86] was shown, which is critical to nervous system partnering.

Such involvement of ECS from the earliest stages of nervous system development is supported by the identification of ECS components, both receptors and enzymes controlling the endocannabinoid metabolism, as well as the endocannabinoids 2-AG and AEA at those stages. Both CB_1_R transcripts and protein were identified in the neural plate, during neurulation and onwards in chick embryos [87,88], as well as 2-AG and AEA and the enzymes involved in their metabolism [88]. CB_1_R expression in such early stages of nervous system development was observed also in zebrafish [89,90] and rodents (from E7.5) [91]. GPR55 mRNA expression was also recently found at such early stages in zebrafish [90].

Particularly concerning the brain, CB_1_R mRNA can be detected in the mice telencephalon both in the pallium and subpalium from E11.5 [91,92,93,94], increasing their expression along with neuronal differentiation [92], as also observed in chick embryos [87,95], peaking at E16.5 [92,96]. Embryonic-derived neural progenitors *in vitro* display functional CB_1_R [97,98,99], and there is some evidence of mRNA expression in proliferative ventricular regions [94], but there is a consistent body of evidence pointing to an absence or very low levels of CB_1_R protein in both ventricular and subventricular zones (VZ/SVZ) in the developing brain [92,93,100,101,102]. CB_1_R immunoreactivity has been identified in intermediate precursor cells exiting the subventricular zone [92], but there is clearly a robust increase in CB_1_R expression in post-mitotic neurons in the developing brain [92,93,100,102,103,104]. A similar pattern of CB_1_R expression in more differentiated cellular stages has been observed in the developing human brain, detected as early as gestational week (GW) 9 [105], and more recently in the monkey, displaying a higher immunoreactivity for CB_1_R in comparison with mice, but completely absent in the VZ/SVZ [102]. Accordingly, in developing cortex, CB_1_R immunoreactivity has been detected in mice at E12.5-E13.5 in the preplate in reelin-expressing Cajal-Retzius cells and newly differentiated glutamatergic neurons [93,100,104], also observed in the developing human brain [105], and later on in post-mitotic radial migrating principal neurons [102,104,106] and migrating interneurons [100,104,107,108]. A similar increase in CB_1_R expression with neural differentiation was also observed in human inducible pluripotent stem cell (IPSC)-derived organoids [109]. From E13.5, CB_1_R expression becomes transiently prominent in developing axons of pyramidal neurons in the intermediate zone (IZ; [104]), in particular in long-range corticofugal axonal tracts such as cortico-thalamic and cortico-spinal tracts [92,93,103,108,110], and perinatally in the afferent fibres cruising the brainstem and cerebellum [111,112]. Such transient prominent subcellular expression in developing axons is also observed in embryonic chicken [87,95], zebrafish [95] and rats [113]. A similar pattern of expression of CB_1_R in neuronal fibre tracts is also observed in the developing human brain [105,114,115,116]. Such transient cellular and subcellular distribution at cortical projection neurons fades in early postnatal life coincident with synaptic contact formation/stabilisation [93,94,110]. CB_1_R is also present in developing cholinergic neurons [117]. Such early expression of CB_1_R indicates that the developing brain is potentially susceptible to exogenous cannabinoids from the very beginning.

The spatial–temporal dynamics in the cellular and subcellular expression of CB_1_R are accompanied by a precise spatial–temporal tone of endocannabinoids (eCB) controlling the activity of CB_1_R tightly regulated by a concomitant dynamic cellular and subcellular distribution of the enzymes controlling the metabolism of eCBs. While in early embryogenesis, AEA seems to take a prominent role [68,96,118], at mid-late embryogenesis, in brain development, 2-AG gains relevance [96,103]. For instance, the existence of a precise and concerted cellular and subcellular expression of DAGL and MAGL has been elegantly shown, supporting a spatially restricted bioavailability of 2-AG necessary for the correct axonal guidance and growth of corticofugal axons [103,108,110,119] and development of cholinergic afferents [117,120] (see below).

CB_2_R has also been identified in embryonic-derived neural progenitors *in vitro* [121,122], supported by the observation of an increase in cell proliferation in E14.5 mice-derived cortical slices upon a selective activation of CB_2_R [98]. Interestingly, in contrast to the finding observed for CB_1_R, its expression decreases with differentiation [121]. Additionally, it has been provided evidence for its expression in retinal ganglion cells that project to the thalamus and midbrain [123] and functional evidence for CB_2_R expression in oligodendrocytes and their progenitors [124].

Regarding the other receptors able to sense eCBs, TRPV_1_ can be transiently expressed during embryonic development in some brain regions [125], and prenatal capsaicin exposure in mice (E7-E13) has a behavioural outcome [126]. Yet, its eventual expression in the developing brain remains elusive. In relation to GPR55, as aforementioned, mRNA expression has been shown throughout the developing brain in zebrafish [90]. Functional evidence suggests its expression in retinal projections [127]. Yet, its presence in the developing brain also remains poorly defined.

### 2.1. Cannabinoid Receptors and the Development of Brain Cytoarchitecture

The development of brain cytoarchitecture encompasses the proliferation and differentiation of neurons and their migration to their final positions in a tightly regulated manner in order to attain subsequent and proper brain wiring. Pharmacological or genetic manipulation of ECS interferes with brain cytoarchitecture in both the number and final position of different neuronal populations from glutamatergic [92,94,102,104,106,109,128] to GABAergic [37,129,130] or cholinergic neurons [117,120]. This may arise from control of proliferation and/or neuronal migration and differentiation by ECS, for which evidence has been provided.

*In vitro* studies in cultured embryonic-derived neural progenitor cells (NPC) indicate that NPCs produce and release the two major eCB species, namely AEA and 2-AG [97], and pharmacological and genetic manipulation (KO mice) of both CB_1_R and CB_2_R showed that the activity of either CB_1_R or CB_2_R promotes proliferation of cultured NPCs derived from different embryonic brain regions [97,98,121,122,131,132]. Accordingly, the increase in the tonic activity of ECS by inhibition or deletion of FAAH induces an increase in NPC proliferation [97]. CB_2_R-induced cell proliferation has also been observed in organotypic E14.5 mice-derived cortical slices [98]. *In vivo*, it has been shown that CB_1_R-KO mice display a reduced proliferation in the developing cortex [92,94,133] (Figure 2), hippocampus [134] and cerebellum [132]. Further evidence indicated that activation of CB_1_R promoted proliferation, inhibiting neuronal differentiation, as observed *in vitro* both in human neural stem cells [135] and cultured embryonic NPC [136]. *In vivo*, CB_1_R was also shown to control the generation of Tbr2^+^ intermediate precursor cells, and its absence (CB_1_R-KO) leads to premature cell cycle exit [101]. This promotion of cell proliferation during development by CB_1_R may entail a bidirectional cross-talk with TNFα [131]. In contrast, WIN55,212-2 exposure during embryogenesis had no effect on cell proliferation [104]. Also, exposure of murine NPCs to AEA has been shown to decrease proliferation [137], and in mouse neural stem cells, activation of CB_1_R favoured differentiation into neurons [99]. In fact, the evidence pointing to an absence or very low levels of CB_1_R in proliferative regions both in the ganglionic eminences in the subpallium [108] and in the developing cortex [102] led to question if the observed CB_1_R-mediated promotion of cell proliferation *in vivo* may be due to a direct action [102]. In addition, it should be noticed that GPR55 activation promotes both proliferation and differentiation of human neural stem cells [138], which needs to be further addressed to better understand the eventual contribution of GPR55 to the role of ECS in neurogenesis.

A more consistent body of evidence supports the involvement of ECS in neuronal migration and differentiation of post-mitotic neurons, in line with the increased expression of CB_1_R along differentiation [87,92], which contributes to the development of cytoarchitecture. Interference with the ECS by prenatal exposure to cannabinoids or genetic manipulation of CB_1_R affects brain cytoarchitecture (e.g., [152,153]), both excitatory [92,94,102,104,106,109,128,154] and inhibitory [40,129,130]. CB_1_R-KO mice display at P2.5 a different distribution pattern of cortical projection neurons labelled with BrdU at E14.5, presenting a higher number of cells at deeper layers and lower at superficial layers [92]. Accordingly, while pharmacological activation of CB_1_R accelerates radial migration, overexpression of the FAAH enzyme inhibits radial migration [92]. Such tonic action of ECS through CB_1_R in radial migration was later reinforced by the observation that the knockdown of CB_1_R at E14.5 in mice by in utero electroporation of plasmids encoding siRNAs induced an accumulation of migrating neurons in the IZ and, consequently, a lower number of cells reaching the cortical plate (CP) at E17.5 [106] (Figure 2). This resulted in an increase in the number of cells at the deeper layers and a decrease in the upper layers at P2 and P10 [106], similar to the observed in the CB_1_R-KO mice [92], indicating a delay in radial migration in the absence or reduced levels of CB_1_R in post-mitotic neurons [106]. Morphological analysis of radially migrating neurons in CB_1_R-KO mice revealed that these neurons at the IZ display deviations in their vertical orientation with misoriented processes, suggesting a role of CB_1_R in correct cell movement from the IZ to the CP [102]. In this regard, it should be mentioned the reported ability of CB_1_R, endogenously activated by 2-AG, to increase neuronal motility of E14.5 mice-derived NPC, increasing the frequency of bursts of movement while reducing their turning frequency [139]. Besides an eventual control of movement, at the IZ, migrating neurons need to polarise, undergoing a multipolar-bipolar transition [155,156], forming a leading process (future apical dendrite) oriented towards the CP and a trailing process (future axon) growing orthogonally to the radial migration direction, in the transition from the lower to the upper IZ [157], necessary for subsequent radial migration towards the CP [158,159], through glial fibre–dependent guidance. While radial glial scaffold seems not to be affected by CB_1_R activation [104], CB_1_R-KO mice at the IZ display a considerably low percentage of cells with a bipolar morphology in comparison with wild-type mice embryos [102]. This indicates that CB_1_R may eventually affect radial migration at the IZ-CP transition by controlling neuronal polarisation and/or through the well-established control of axon formation/outgrowth (see next section; Figure 2). In addition, by controlling the neuronal differentiation of glutamatergic neurons [92], CB_1_R is also involved in cortical projection neuron distribution across the different cortical layers, in particular by controlling the differentiation/maturation of deep cortical layer 5 pyramidal neurons [94,109,128]. While genetic ablation of CB_1_R in post-mitotic cortical projection neurons reduced the number of sub-cerebral projection neurons of layer 5 (Ctip2^+^) and a consequent decrease in cortical thickness, FAAH-KO mice displayed a higher number of Ctip2^+^ cells [94]. Interestingly, interfering with this tonic action induced by 2-AG via CB_1_R by prenatal exposure to Δ^9^-THC in mice between E12.5-E16.5 also reduced the number of neurons in layer 5 [128]. Such balanced CB_1_R activity drives the generation of deep-layer Ctip2^+^-neurons by preventing Satb2-mediated repression, increasing Ctip2 expression [94] (Figure 2). These observations performed in mice were more recently recapitulated in human IPSC-derived brain organoids [109]. The dysregulation of CB_1_R-mediated generation of sub-cerebral projection neurons leads to long-term impairments in corticospinal motor function [94,128]. ECS may also affect early-born cortical projection neurons placement through CB_1_R expressed in Cajal-Retzius cells [93,100,104], which contributes to guiding early-born post-mitotic glutamatergic neurons through the expression of reelin [157,160,161] since CB_1_R controls the number of Cajal-Retzius cells [104] (Figure 2).

As already mentioned, ECS also controls the development of inhibitory cytoarchitecture since prenatal Δ^9^-THC exposure or genetic deletion of CB_1_R (KO mice) affects the number of different types of interneurons [40,129,130]. This seems to reflect, on the one hand, a CB_1_R-mediated control of tangential migration of interneurons since WIN55,212,2 exposure from E5 in rats induced an increase in the number of GABA cells tangentially migrating in the marginal zone [104]. This should entail a chemoattract action of ECS through CB_1_R activation on migrating interneurons, as chemotaxis of CCK^+^-interneurons by CB_1_R was observed *in vitro* through the transactivation of TrkB receptors [40], previously shown to be involved in the tangential migration of medial ganglionic eminence-derived cells [162] (Figure 2). Such interplay between ECS and BDNF may also be involved in radial migration (see [163]). Moreover, there is also the functional interplay between neuregulin-1, which is a major chemoattractant of cortical tangentially migrating interneurons [164] and ECS. Neuregulin-1 downregulates MAGL expression, leading to enhanced 2-AG signalling [165], and a cross-talk between neuregulin-1 and ECS was observed in the control of movement of cortical embryonic neuroblasts [139]. This opens the possibility of ECS also controlling the guidance/movement in tangential migration through interaction with neuregulin-1. Moreover, while principal neurons are endowed with the capacity of eCB synthesis during their development, self-sustaining ECS [92], GABAergic interneurons seem to lack synthetic enzymes until the switch to radial intracortical migration [108], most likely attracted by paracrine guidance by target-derived eCBs. This suggests that ECS may play a role in the integration of the excitatory and inhibitory cytoarchitecture. Furthermore, ECS is also involved in the differentiation/maturation of GABAergic neurons, also shown to involve the activity of TrkB receptors [40].

In addition to the control of differentiation of glutamatergic and GABAergic neurons, ECS also contributes to the differentiation of cholinergic neurons [117,120]. Cell-autonomous DAGLα-derived 2-AG signalling via CB_1_R controls the spatial organisation and morphogenesis of cholinergic neurons with an impact on cholinergic basal forebrain projections. This is under the control of nerve growth factor (NGF) through TrkA receptors by regulating 2-AG spatial availability through the control of MAGL subcellular levels [117,120].

ECS is also involved in gliogenesis. In cultured neuronal progenitor cells derived from P2 rat cortices, the pharmacological activation of CB_1_R increased the generation of GFAP^+^ cells [134]. *In vivo*, CB_1_R-KO mice displayed a decrease in astrogliogenesis and an increase in neurogenesis in rat developing hippocampus postnatally (P15), in contrast to a CB_1_R-induced neuronal commitment observed prenatally (e.g., [92,94,99]). A more consistent body of evidence supports the role of ECS in oligodendrogenesis. 2-AG produced by cultured rat-derived oligodendrocyte precursors (OPC) expressing DAGLα and DAGL [166], and CB_1_R and CB_2_R [158], promoted both OPC survival [140], proliferation [141] and oligodendrocytes differentiation [166] via CB_1_R or CB_2_R, through PI3K/Akt and mTOR signalling [140,141,142] (Figure 2). *In vivo*, while the postnatal (P1-P15) activation of CB_1_R in rats induced an increase in oligodendrocyte cell commitment, CB_2_R was more associated with migrating OPCs [143]. Yet, only the activation of both CB_1_R and CB_2_R increased the expression of myelin basic protein in subcortical white matter [143]. Accordingly, postnatal Δ^9^-THC exposure (P6-P9) in mice increases the density of mature myelinating oligodendrocytes in subcortical white matter, decreasing OPC by inducing OPC cell cycle exit while promoting oligodendrocyte differentiation, effects prevented by selective antagonists of CB_1_R or CB_2_R [167]. Likewise, in mice, at late embryogenesis, ECS also promotes oligodendrocyte differentiation since the inhibition of MAGL *in vivo* leads to premature differentiation of oligodendrocytes, although only via CB_2_R and not CB_1_R [124] (Figure 2).

### 2.2. Cannabinoid Receptors and the Development of Brain Circuitry

ECS is involved in the development of brain circuitry, not only by governing the development of cytoarchitecture but also through involvement in the axonal pathfinding for the formation of synaptic connectivity and their maturation/refinement. 

As mentioned, during brain development, CB_1_Rs display a predominant expression in developing axons [168], particularly in distal segments and growth cones, as observed in diverse neuronal types such as glutamatergic [92,93,94,103,108,110,144], GABAergic [108] or cholinergic neurons [117,120]. Activated by target-derived 2-AG or produced by DAGL located in the axonal tips [92,103,108,110,120,123] and spatially limited to the motile growth cones by MAGL located at proximal axonal segments [103,117,120], CB_1_R promotes axonal development by controlling their directional growth [95,103,110,117,119,120,124,169]. This is achieved by a chemorepulsion action of CB_1_R at the actin-rich growth cone, including motile filopodial extensions, driving growth cone steering [103,108,117,124,144]. Accordingly, the genetic or pharmacological manipulation of CB_1_R, DAGL or MAGL has been shown to have an impact on the development of axons and correct axon pathfinding from diverse neuronal populations. 

In developing chick embryos or zebrafish, the genetic knockdown or pharmacological blockade of CB_1_R impairs axonal growth, guidance and fasciculation [95,170]. In mammals, in agreement with the observed transient expression of CB_1_R in white matter tracts in long-range corticofugal developing axons at mid-late embryogenesis [92,93,100,108,113,115,116,171], the genetic deletion of CB_1_R selectively in post-mitotic cortical projection neurons impaired axon fasciculation of corticofugal axons due to impaired axon pathfinding [92], both corticothalamic [110] or corticofugal tracts [94] (Figure 2). Interestingly, CB_1_R-KO mice display aberrant fasciculation and misrouting not only of corticothalamic axons (CTA) but also of thalamocortical axons (TCA) [110]. Taking into account that CTAs express CB_1_R, whereas TCAs do not, but express MAGL and DAGL [103,110], these findings indicate that CB_1_R signalling in CTA, triggered by tightly spatially-regulated availability of 2-AG, is involved not only in the development of CTA but also in the partnering of TCA, mediating the reciprocal fasciculation of afferent and efferent cortico-thalamic projections [103,110]. Indeed, CB_1_R-KO mice display a significant increase in the innervation by thalamocortical axons of cortical layers 2/3 [145], although it may also entail activity-dependent mechanisms (see below). In agreement with the role of CB_1_R in the development of long-range axonal projections, in utero exposure to Δ^9^-THC also impairs corticofugal tracts [128,154]. In contrast, MAGL inhibition triggered corpus callosum enlargement due to corticofugal axon spreading [124]. Moreover, it was elegantly shown that ECS guides corticofugal axons by a concomitant CB_1_R-induced Robo1 positioning at the growth cones and a CB_2_R-induced production of Slit2 by oligodendrocytes inducing a chemorepellent signal [124] (Figure 2). Concomitantly, it may be involved in the myelination of these fibres, as Δ^9^-THC exposure enhanced subcortical white matter myelination in a CB_1_R and CB_2_R-dependent manner [167].

The development of retinal projections in mice also entails CB_1_R-driven guidance by controlling growth cone steering [144]. This was shown to be mediated by the regulation of the trafficking of deleted in colorectal cancer (DCC) receptors, which tethers netrin-induced growth cone steering in a PKA-dependent manner [144]. A similar mechanism was observed in cultured cortical neurons [161]. *In vivo*, while the activation of CB_1_R reduced retinal projection growth, its blockade promoted growth and caused aberrant projections [144]. Later on, it was shown that CB_2_R is also expressed in retinal ganglion cell growth cones and engaged in the development of their axons by controlling growth cone morphological changes through a similar mechanism [123]. Likewise, genetic deletion or pharmacological blockade of CB_2_R increased retinal axonal length and aberrant projections, affecting retino-thalamic projections [123]. GPR55 was also shown to regulate retinal axon growth and guidance. Pharmacological activation of GPR55 increases the surface area and filopodia in growth cones, inducing retinal axon growth [127]. *In vivo*, GPR55 activation leads to aberrant retinal ganglion cell projections affecting target selection [127].

ECS also controls axon pathfinding of cortical GABAergic interneurons through CB_1_R [108]. While eCBs were shown to be chemoattractants in interneuron migration [40], they control their axonal guidance by inducing growth cone collapse through CB_1_R activation, most likely by a target-derived 2-AG, as suggested by downregulation of DGAL with GABAergic differentiation [172] and by the observed dendritic redistribution of DAGL in glutamatergic pyramidal cells at late embryogenesis [108]. CB_1_R also controls cholinergic innervation of the hippocampus. CB_1_R activated by cell-autonomous 2-AG signalling produced by DAGL, co-located with CB_1_R at the growth cones and spatially restricted to the motile segments by MAGL selectively located at the proximal axonal stems, facilitates the outgrowth of cholinergic afferents, inhibiting growth cone differentiation, while controlling their guidance, eventually by 2-AG paracrine signalling [117,120]. This role of CB_1_R signalling in cholinergic axon pathfinding was shown to be regulated by NGF [120]. More recently, the observation that CB_1_R-KO-mice display impaired striatonigral connectivity suggests a role of CB_1_R also in axonal pathfinding of striatal neurons onto dopaminergic neurons in the substantia nigra [173]. Concerning the TRPV_1_ receptor, the observation that temperature-induced axonal repulsion in rat cortical neurons is mediated by TRPV_1_ [174] suggests that it may also be involved in axonal pathfinding, yet its role remains ill-defined.

The chemorepellent signalling induced by CB_1_R and controlling axon guidance was first shown to involve RhoA activation and subsequent ROCK activation in GABAergic interneurons [108] (Figure 2). In cultured rat hippocampal neurons and organotypic slices, this was shown to induce non-muscle myosin II-dependent contraction of the actomyosin cytoskeleton, leading to actin-rich growth cone retraction, a mechanism shown to be required for the correct pathfinding of corticofugal neurons [175]. Moreover, in mice developing cortical neurons, CB_1_R-induced growth cone collapse was shown to entail a deactivation of Rac1 leading to F-actin disassembly, being proposed that CB_1_R induces the retraction of filopodia by Rac1 deactivation and of lammelipodia by RhoA activation [176]. In fact, both DCC trafficking and the Slit-Robo pathway shown to be involved in CB_1_R-mediated growth cone repulsion in retinal ganglion cells and cortical neurons [124,144] have been associated with RhoA [177] or Rac [178]. Hence, similar intracellular mechanisms seem to be engaged by CB_1_R to induce growth cone collapse in different neuronal populations. Concomitantly, CB_1_R may also be able to control microtubule stability by regulating superior cervical ganglion 10 (SCG10)/stathmin-2 protein [154], involved in microtubule disassembly [179]. Furthermore, the targeting of CB_1_R to axonal growth cones, namely in corticofugal axons, was recently shown to be mediated by kinesin-1 [180]. The genetic deletion of kinesin-1 leads to abnormal fasciculation and pathfinding defects of corticofugal axons with a reduction in CB_1_R levels [180]. When the axon reaches its postsynaptic target, there is a cellular and subcellular redistribution of the ECS components. Essentially, MAGL accumulates in growth cones, limiting 2-AG signalling, most likely decreasing the growth cone motility, allowing presynapse differentiation, and keeping a presynaptic location [103], whereas DAGL is targeted to postsynaptic dendritic spines [92,108,119] for retrograde signalling.

Regarding synaptogenesis *per se*, in cultured rat hippocampal neurons, the pharmacological activation of CB_1_R inhibited synapse formation [181]. The inhibition of tonic activity of CB_1_R by DAGL inhibition induced an increase in synaptogenesis in cultured cortical neurons [92]. An increase in synaptogenesis with the antagonism of CB_1_R was also observed in a cortical spheroid model of human brain development [182]. *In vivo*, the genetic deletion of CB_1_R in cortical interneurons led not only to an increase of inhibitory synaptic contacts at cortical pyramidal cells but also to an altered synaptic distribution [108]. Likewise, genetic deletion of DAGLα impairs cholinergic afferents in the hippocampus, but mainly their targeting and not their density [120]. Hence, ECS and CB_1_R seem to contribute to the development of synaptic contacts mainly through the morphological guidance towards their postsynaptic target rather than a direct role in the structural formation of synapses. In spite of this, recent evidence indicates that CB_1_R can contribute to synapse formation and stabilisation, but in an activity-dependent manner [183]. In mice organotypic slices, it was shown that exogenous activation of CB_1_R induces the formation and stabilisation of inhibitory boutons at principal neurons, independently of neuronal activity [183]. However, physiologically, this is triggered in locations of strong excitatory input, entailing postsynaptic 2-AG production and activation of CB_1_R at inhibitory axons, most likely to tune excitation/inhibition balance [183,184]. Furthermore, ECS through CB_1_R is also involved in synaptic circuitry refinement in an activity-dependent manner. CB_1_R-KO mice display altered circuitry in the primary somatosensory cortex [153,185] and visual cortex [152], most likely due to the deletion of CB_1_R at glutamatergic neurons [153]. This may reflect in part the role of CB_1_R in the development of cytoarchitecture and axon pathfinding. However, it seems also to rely on the control of synaptic pruning by CB_1_R through the induction of long-term depression (LTD) (Figure 2), which is a key activity-dependent process in neuronal circuitry refinement by selective elimination of redundant or weak synaptic connections [186]. In the mouse visual cortex, the blockade of CB_1_R during brief monocular deprivation prevented experience-dependent synaptic weakening selectively at L2/3 by blocking CB_1_R-induced LTD [146]. Likewise, in rodent primary somatosensory cortex, CB_1_R-LTD is also required not only for the weakening of deprived sensory inputs in L2/3 but also of L4-L2/3 synapses [147], previously shown to display a CB_1_R-dependent LTD [148,149] (Figure 2). This may contribute to normal circuit development since CB_1_R blockade disturbed whisker map formation [147]. CB_1_R expressed in TCA-L2/3 synapses controls their synaptic pruning through the ability to induce LTD [145]. CB_1_R also seems to control the pruning of glutamatergic synapses and eCB-mediated LTD in rat prefrontal cortex (PFC) [150,151]. CB_1_R may also contribute to synaptic pruning by mediating hetero-LTD as observed in L2/3 of the mice visual cortex [187] and developing hippocampal CA1 area in rats [188] in the first two postnatal weeks. CB_1_R may also interfere with circuitry development and maturation by controlling the excitatory–inhibitory switch of GABAergic signalling since Δ^9^-THC exposure in postnatal days 1–10 caused a delay in this switch via CB_1_R [189]. This control of synaptic pruning/refinement by ECS constitutes a particular window of vulnerability for exogenous cannabinoids, especially during adolescence, with possible lasting consequences (see next section).

In addition to glutamatergic, GABAergic or cholinergic signalling, ECS may also be involved in the development of other neurotransmitter signalling systems, as suggested by studies showing that the exposure to cannabinoids perinatally can also affect, for instance, dopaminergic (e.g., [190,191,192]) or serotoninergic [193,194] systems.

## 3. Cannabinoids and the Adolescent Brain

Adolescence represents a period of profound neurodevelopment, marked by structural and functional changes within the brain’s cytoarchitecture and synaptic circuitry, particularly in the PFC, a region critical for executive functions, decision-making, and impulse control [195,196]. This transition from childhood to adulthood involves the maturation of several brain regions, particularly the PFC, amygdala, and hippocampus, which regulate executive functions, emotions, and learning. These areas undergo extensive synaptic pruning, myelination and circuit refinement, making adolescence a sensitive window for both adaptive and maladaptive plasticity [197,198]. These fine-tuning processes eliminate redundant or weak synapses and facilitate signal transmission across brain circuits, especially those related to cognitive and emotional regulation [199]. Human imaging studies reveal significant reductions in grey matter volume in the PFC and temporal lobes during adolescence, consistent with synaptic pruning observed in animal models [200,201]. White matter increases, attributed to enhanced myelination, have also been documented in regions such as the corpus callosum and other subcortical areas [202,203]. These structural changes reflect a shift toward more efficient neural processing and enhanced cognitive control, with notable improvements in functions such as working memory, impulse control, and decision-making [204,205]. However, this ongoing synaptic and circuit refinement opens a critical window during which external factors such as substance use can significantly influence brain development [199].

Adolescence also coincides with increased risk-taking behaviours, emotional instability, and heightened social influence, potentially leading to drug experimentation, including cannabis use [196,206]. The developing brain is particularly vulnerable to cannabis exposure, which has been associated with various negative outcomes, including impaired cognitive function, increased risk of psychiatric disorders, and long-lasting changes in brain structure [207,208]. Adolescence is a critical period when both the dopaminergic system and the ECS take centre stage in PFC development [209]. The susceptibility of the adolescent brain to such effects is thought to stem from the intricate roles of the ECS in the ongoing brain maturation. During adolescence, the ECS undergoes dynamic changes, with peaks in CB_1_R expression and endocannabinoid ligand levels observed in the PFC and hippocampus [210,211,212,213] (Table 1). These fluctuations make the adolescent brain highly sensitive to perturbations in ECS, including those induced by exogenous cannabinoids such as Δ^9^-THC [208]. Both human and animal research demonstrate that adolescent cannabis exposure results in persistent changes to brain structure, function, and behaviour. These changes increase the risk of psychiatric disorders, including anxiety, depression, and schizophrenia, and result from the disruption of normal ECS during a critical period of brain development. Understanding how cannabis use during adolescence affects the maturation of the ECS and related neural circuits is critical for developing interventions to mitigate its long-term consequences [214,215].

### 3.1. Animal Studies on the Role of the Endocannabinoid System in the Adolescent Brain

Adolescence is also a crucial period for rodent brain development, characterised by dynamic changes in corticolimbic structures [240]. These regions, including the PFC, amygdala, and hippocampus, are involved in regulating emotional behaviours such as fear, anxiety, and executive function. The ECS plays a central role in controlling the orchestration and the function of these circuits, primarily through the CB_1_R [222]. Rodent studies have revealed that the ECS undergoes significant developmental changes during adolescence. The ECS regulates the balance between excitatory and inhibitory neurotransmission, which is crucial for the maturation of synaptic connections and the refinement of corticolimbic circuits [14,222]. The expression of CB_1_Rs peaks at the onset of adolescence, especially in the PFC and striatum, before declining into adulthood [241]. In adolescent rats, Molla et al. (2024) found that the ECS was not yet fully engaged to regulate afferent transmission from these brain regions [242]. By late adolescence, however, both 2-AG and anandamide could be recruited to limit hippocampal drive, although only 2-AG inhibited basolateral amygdalar inputs. The protracted development of the ECS in the PFC and its fluctuating developmental trajectory in other corticolimbic regions may leave the adolescent brain particularly vulnerable to disruptions by cannabis exposure during this critical window of development [210,242].

These vulnerabilities can be assessed in adolescent rodents exposed to cannabinoids, as this experimental paradigm recapitulates key behavioural and structural alterations that are often found in regular cannabis consumer adolescents [214,215,222]. The following animal studies unanimously indicate that perturbations in ECS signalling during adolescence, whether through stress or exogenous cannabinoid exposure, can result in long-lasting effects on emotional regulation and cognitive processing [243]. In the rodent brain, significant cellular and molecular alterations can be found after cannabinoid exposure, particularly in the PFC, hippocampus, and other corticolimbic areas. Importantly, these are brain areas critical for memory and cognition. Chronic exposure of adolescent rodents to Δ^9^-THC or synthetic CB_1_R agonists has been shown several times to cause long-term impairments in tasks such as short-term memory, object recognition, spatial working memory, social interaction memory, and affective functions [214,244]. These effects are associated with changes in proteins involved in synaptic plasticity (e.g., PSD95, NMDA receptors), abnormal firing patterns of pyramidal neurons, reduced dendritic complexity, especially of the pyramidal neurons in layer 2/3 in the medial PFC (mPFC) and reduced hippocampal connectivity, together with the downregulation and desensitisation of CB_1_Rs in various brain regions, with a more pronounced effect in females. This is likely due to dynamic and sexually dimorphic changes in the expression and molecular pharmacology of CB_1_Rs during adolescence, especially in regions involved in cognition and emotional regulation [215,222].

Indeed, Bernabeu et al. (2023) reported how synaptic plasticity, particularly eCB-LTD, exhibits sex-specific differences during adolescence [216]. While other forms of plasticity, like long-term potentiation (LTP) and mGluR-LTD, are already mature in both sexes by adolescence, eCB-LTD is expressed early in females but only appears at puberty in males. This study also found greater synaptic levels of CRIP1a (a CB_1_R-interacting protein that reduces CB_1_R signalling via G proteins) and ABHD6 in juvenile males, which likely contributed to the repressed eCB signalling as compared to juvenile females. Additionally, this milestone study systematically analysed the expression of other elements of the eCB system across both sexes of juvenile, pubescent and adult rats, and they found significant and likely meaningful age- and sex-dependent changes in the expression of the CB_1_R, CB_2_R, TRPV_1_R, DAGLα, MAGL, NAPE-PLD, FAAH and mGluR5 (the activity of the latter is associated with retrograde 2-AG release; see above). These findings highlight that synaptic plasticity in the PFC is not uniform across sexes or developmental stages. The differences were specific to the PFC and were not observed in other brain regions like the nucleus accumbens, supporting the notion that the PFC is one of the last regions to mature (Table 1).

In conclusion, the findings of Bernabeu et al. (2023) underscore the critical role of the ECS in adolescent brain development and the long-term impacts of early cannabinoid exposure [216]. Adolescence is a period of heightened vulnerability to changes in synaptic plasticity, and sex-specific differences in ECS function may shape how the brain responds to cannabinoid agonists during this crucial developmental window. In line with this affirmation, adolescent rodents exposed to cannabinoids showed impaired maturation of the glutamatergic and GABAergic systems, in particular, abnormal glutamate receptor distribution and altered inhibitory/excitatory balance. At the ultrastructural level, disrupted normal patterns of synaptic pruning, reduced dendritic spine density and alterations in dendritic length and remodelling were observed in the hippocampus and PFC of adolescent rodents subject to cannabinoid agonist exposure [215,222].

Adolescence can be divided into early, mid-, and late stages, with cannabinoids potentially exerting distinct effects on synaptic circuit maturation in the PFC across these phases. Rubino and colleagues (2015) exposed female rats to Δ^9^-THC from mid-adolescence (35 PND) to late adolescence (45 PND), extending observations into young adulthood (75 PND) [150]. Their study revealed natural developmental fluctuations in CB_1_R, anandamide, and 2-AG levels in the PFC of these female rats. However, Δ^9^-THC exposure disrupted these processes, resulting in reduced CB_1_R density and anandamide levels in adulthood, ultimately impairing eCB-LTD in the PFC. Adolescent Δ^9^-THC exposure also significantly decreased spine density in distal basal dendrites despite an overall increase in PSD-95 protein levels. Furthermore, Δ^9^-THC exposure prematurely disrupted glutamatergic synaptic pruning, as it prevented the normal decline of NMDA receptor GluN2B subunits and the corresponding rise in GluN2A, leading to persistently elevated GluN2B levels in adulthood. Δ^9^-THC exposure also increased AMPA receptor GluA1 subunits without affecting GluA2, suggesting a shift toward calcium-permeable AMPA receptors, which are associated with immature synaptic states and linked to psychiatric vulnerability. As adults, these female rats showed cognitive deficits, particularly in spatial working memory. Notably, blocking CB_1_Rs with the antagonist AM251 from early to late adolescence similarly disrupted glutamatergic maturation, preventing decreases in postsynaptic markers such as PSD-95, GluN2A, and GluA2, which typically facilitate synaptic refinement and pruning [150].

Interestingly, another study found significant differences in the effects of chronic Δ^9^-THC treatment in late adolescent male rats. Miller et al. (2019) confirmed that synaptic pruning is a crucial developmental process in the PFC from late adolescence to early adulthood in rats, marked by expansions in basal dendritic arborisation and dendritic spine pruning that contribute to circuit refinement [151]. However, Δ^9^-THC exposure in these late-adolescent males disrupted typical PFC maturation by prematurely inducing spine pruning and causing allostatic atrophy of dendritic arborisation. This abnormal pruning particularly affected layer 3 pyramidal neurons, key components of PFC circuitry with connections to brain regions like the amygdala and thalamus. Additionally, Δ^9^-THC exposure resulted in a distinct transcriptomic profile, with significant changes in genes related to chromatin remodelling and histone modification, showing minimal overlap with control rats. The disrupted gene networks primarily involved cell morphogenesis, dendritic development, and cytoskeletal organisation. Notably, dysregulated gene expression in Δ^9^-THC-exposed rats paralleled patterns observed in schizophrenia patients, particularly in genes linked to cytoskeletal and neurite development. Altogether, this study underscores that adolescent Δ^9^-THC exposure can significantly alter the developmental trajectory of the PFC by modifying both neuronal structure and gene expression, potentially increasing susceptibility to psychiatric disorders such as schizophrenia [151]. These findings highlight the critical nature of the late-adolescent period for PFC development and its vulnerability to exogenous cannabinoids.

Synaptic maturation depends not only on synaptic activity but also on intact glial cell function. Exposure to cannabinoid agonists during adolescence can further modulate the function of various glial cell types. There are several studies reporting changes in astrocytic markers (GFAP) and microglial morphology, contributing to neuroinflammation and abnormal synaptic pruning during brain maturation. These alterations lead to worsened working memory, cognitive flexibility and spatial recognition tasks, which is translated into persistent impairments in executive functions and decision-making [214,215] (Table 1). The role of microglia in adolescent brain development is far from fully appreciated. CB_1_R expression in microglial cells is often overlooked due to levels being an order of magnitude lower compared to CB_1_R expression in neurons [34].

This recent study by Hasegawa et al. (2023) examined the interaction between adolescent Δ^9^-THC exposure and genetic predisposition to psychiatric disorders, modelled using a 16p11.2 duplication (16p11dup) in mice [34]. Preclinical studies have shown that the 16p11dup mouse model displays cognitive behavioural abnormalities, along with structural irregularities in the dendrites of pyramidal neurons and GABAergic synapses in the prefrontal cortex [34]. Adolescent Δ^9^-THC treatment resulted in a significant decrease in microglial cellular processes, an increase in microglial cell body size, and accelerated microglial apoptosis due to upregulated p53 signalling in the mPFC of male mice. These changes were not observed in female mice. The effects of Δ^9^-THC were more pronounced in 16p11dup mice, highlighting a gene–environment interaction. The combination of Δ^9^-THC exposure and 16p11dup led to a synergistic increase in microglial apoptosis and a reduction of Iba1^+^ microglia, specifically in the mPFC. This interaction also exacerbated deficits in social memory. Functionally and behaviourally, these alterations in Δ^9^-THC-treated adolescent 16p11dup male mice resulted in reduced excitability of pyramidal-tract neurons in the mPFC and impaired social memory in adulthood. Notably, microglia-selective deletion of CB_1_R prevented the changes induced by adolescent Δ^9^-THC exposure, underscoring the critical role of microglial CB_1_R in mediating the adverse cognitive and social effects of adolescent Δ^9^-THC exposure, particularly in genetically predisposed individuals [34].

Lee et al. (2022) also examined the effects of adolescent low-dose Δ^9^-THC exposure on microglial function and the broader ECS, particularly focusing on how Δ^9^-THC disrupts microglia’s homeostasis and impairs their responses to microbial infection and social stress into young adulthood [218]. Repeated low-dose Δ^9^-THC exposure during adolescence induced a state of dyshomeostasis in microglia isolated from the brains of male and female mice. This was evident from broad alterations in the expression of genes critical to microglial homeostasis, such as those related to innate immunity (e.g., Il-1β, Il-6, Tlr2-9). The observed dysfunction persisted into early adulthood (postnatal day 70) but returned to baseline at full maturity (postnatal day 120), thus revealing a critical period in adolescence where Δ^9^-THC can significantly disrupt microglial function, which in turn could influence brain health during crucial developmental windows. The study of Lee et al. (2022) also showed alterations in the ECS upon repeated Δ^9^-THC exposure, particularly in microglial cells [218] (Table 1). This includes an increase in FAAH and a decrease in NAPE-PLD and MAGL expressions. These perturbations imply an enduring change in anandamide and 2-AG signalling, contributing toward the altered immune response and microglial dysregulation. In addition to immune dysregulation, adolescent Δ^9^-THC exposure caused impairments in the response to psychosocial stress (social defeat paradigm). Normally, social stress would induce anxiety-like behaviours and an immune response, but Δ^9^-THC-exposed mice showed a blunted response, suggesting a diminished capacity to handle stress. This further points to long-term effects on the brain’s neuroimmune interface and stress-processing pathways. As already expected from the above studies, sex differences were also observed because male mice showed more pronounced changes in microglial morphology, while both sexes exhibited reduced cytokine responses post-Δ^9^-THC exposure. Surprisingly, these pathological changes were fully abolished by peripheral CB_1_R blockade, suggesting that peripheral CB_1_Rs, potentially on circulating monocytes, may play a key role in mediating Δ^9^-THC’s impact on microglia, highlighting a potential cross-talk between the central and peripheral immune systems [218] (Table 1). 

However, the impact of cannabinoid agonists on microglia, especially those that are selective for the CB_2_R, can be positive, too. For instance, it is known that chronic alcohol exposure (CAE) during late adolescence increases anxiety-like behaviours, especially during withdrawal, which may persist into adulthood. These effects are linked to neuroinflammation in the PFC. Li et al. (2023) found that CAE triggers the activation of microglia, which displayed deramification (retraction of their processes) and cell body enlargement [219]. These changes are often linked to a transition from a homeostatic (M2-like) to a pro-inflammatory (M1-like) state, which is characterised by the secretion of pro-inflammatory cytokines like IL-1β and TNF-α. These cytokines are involved in synaptic pruning and may damage neuronal circuitry. The authors also found that CAE increased CB_2_R density in PFC microglia, and CB_2_R activation by its selective agonist AM1241 that does not bind CB_1_R prevented CAE-induced anxiety-like behaviours, mitigated microglial activation by reducing their pro-inflammatory M1-like phenotype, restored normal microglial morphology and reduced the secretion of inflammatory cytokines [219]. It suppressed NLRP3 inflammasome activation, which is critical in promoting inflammation through the caspase-1/IL-1β pathway. Altogether, these findings suggest that CB_2_R activation offers a potential therapeutic strategy for treating alcohol-induced neuroinflammation and related mood disorders such as anxiety in late adolescence (Table 1).

Exposure to alcohol and stress is increased during adolescence in many human societies and often negatively impacts brain development in synergism [196]. A recent investigation shed light on the role of hippocampal CB_1_R in impulsivity and alcohol abuse during adolescence [245]. This report demonstrates that adolescent rats exhibit more impulsive choices and consume more alcohol than adults—behaviours that are associated with elevated CB_1_R expression in the CA3 and dentate gyrus (DG) regions of the adolescent hippocampus. These findings support the notion that CB_1_Rs in this brain area play a significant role in mediating impulsive behaviours and substance-seeking tendencies, further emphasising the involvement of ECS in adolescent brain maturation. Besides the CB_1_R, the role of TRPV_1_Rs in mediating stress responses is also implicated in adolescence, suggesting that ECS dysregulation during this critical period may lead to long-term vulnerability to stress-related disorders [216]. In concert with this, another study in adolescent mice found that CAE impairs CB_1_R-dependent synaptic plasticity (eCB-LTD) in the DG medial perforant pathway (MPP-LTD) [220]. Furthermore, environmental enrichment (EE) rescued eCB-LTD, and additionally, in the control mice, EE reverted the eCB-LTD into a novel form of TRPV_1_R-dependent LTP (MPP-LTD to MPP-LTP switch). In conclusion, the study provides evidence that EE influences different synaptic plasticity pathways involving the CB_1_R and the TRPV_1_R in the hippocampus, potentially offering therapeutic strategies to counteract the cognitive deficits induced by adolescent alcohol exposure [220] (Table 1).

Actually, the CB_1_R and the TRPV_1_R have been demonstrated to exert opposing effects on anxiety, the former being anxiolytic and the latter anxiogenic [221]. Hence, simultaneous blockade of FAAH and TRPV_1_R blockade may be an interesting tool to be explored in anxiety disorder in adolescents. Nevertheless, stress, fear and anxiety-related behaviours are difficult to dissociate from one another in animal models, where they have been shown to be particularly sensitive to CB_1_R modulation during adolescence [246,247]. In animal models, cannabinoid exposure produces mixed outcomes regarding anxiety, with some studies reporting anxiolytic effects while others show increased anxiety. CB_1_R activation has been shown to reduce fear and anxiety responses by dampening excitatory inputs in the PFC and amygdala, thereby promoting emotional regulation [248]. Others showed enduring increases in anxiety-like behaviours and dysregulation of the hypothalamic–pituitary–adrenal (HPA) axis in adulthood [222,249] (Table 1).

Disruption of ECS during adolescence also impairs the maturation of fear extinction circuits, leading to persistent deficits in the ability to regulate anxiety and fear responses in adulthood [243,250]. Such findings underscore the importance of the ECS in modulating brain plasticity and emotional development during this critical period. Chronic Δ^9^-THC exposure in adolescent rats reduced dendritic complexity and synaptic density, especially in regions associated with executive function and emotional regulation [215]. This reduction in synaptic strength is accompanied by behavioural deficits, such as increased impulsivity and impaired decision-making [242]. Δ^9^-THC exposure during adolescence has also been associated with depressive-like behaviours, including passive coping strategies and anhedonia. Additionally, adolescent exposure to natural and synthetic cannabinoids affects the mesolimbic dopamine system, probably due to the presence of cannabinoid receptors in both dopaminergic cells and their input terminals [10], further exacerbating decision-making impairments [209,244,251].

One might wonder not only whether chronic alterations in ECS signalling during adolescence shape stress- and anxiety-related behaviours later in life but also whether stress itself influences the ECS in the adolescent brain, creating a reciprocal relationship between stress exposure and ECS modulation during this critical developmental period. Indeed, Demaili and colleagues (2023) recently reported that early life stress (ELS) and adolescent stress independently or in combination influence the ECS of young female rats, particularly the expression of CB_1_R and FAAH in the mPFC [252]. These changes were driven by epigenetic mechanisms, specifically DNA methylation, which led to long-term modulation of stress responses. The findings offer insights into how ELS can reprogramthe ECS to either buffer or exacerbate responses to subsequent stress in adolescence, with implications for mental health outcomes later in life. Curiously, both ELS and adolescent stress independently led to CB_1_R upregulation in the mPFC, suggesting that ECS changes persist into adulthood. However, when ELS was followed by adolescent stress, CB_1_R expression returned to control levels, indicating a “buffering” effect. In contrast, only adolescent stress (forced swimming) caused an upregulation of FAAH, while ELS alone did not have this effect. Nevertheless, ELS exposure buffered the upregulation of FAAH by adolescent stress. These changes in gene expression were paralleled by decreased DNA methylation across specific CpG sites at the promoter regions of the CB_1_R and FAAH genes. Overall, the study supports the two-hit hypothesis, where ELS reprograms the response to later (adolescent) stressors [252] (Table 1).

Altogether, prolonged exposure to Δ^9^-THC or synthetic cannabinoids during adolescence is associated with persistent behavioural abnormalities, such as deficits in social interaction and various types of memory, increased anxiety, anhedonia, and cognitive filtering, which all persist into adulthood. At the neurophysiological level, GABAergic hypofunction is found in the PFC, which contributes to the overactivation of the mesolimbic dopamine system. Furthermore, dysregulation of cortical pyramidal neurons, the reduction in gamma oscillations and sensorimotor gating deficits (prepulse inhibition) are consistently observed in these animal models. At the molecular level, reduced expression of GAD67 and GAT-1 is found, together with dampened signalling pathways such as Akt1/GSK-3 and mTOR, which are associated with the regulation of dopamine and GABAergic neurotransmission [253,254]. Importantly, these alterations strongly resemble schizophrenia-related psychopathology and recapitulate psychosis-related behaviours in men, which is often associated with precedent marijuana use during adolescence (see below) [222,223,224,255] (Table 1).

Recently, a ground-breaking study recapitulated how chronic adolescent Δ^9^-THC exposure leads to severe behavioural, anatomical, and molecular impairments in animals, resembling neuropsychiatric disorders like schizophrenia [244]. The authors used a Δ^9^-THC dosing range that mimics the effects of a moderate to heavy use regimen of marijuana on a human adolescent, and it was previously shown to cause a profound and enduring neuropsychiatric phenotype [253]. As many times seen before and discussed above, these rats display cognitive deficits, affective abnormalities, impaired sensorimotor filtering, aberrant pyramidal cell firing patterns and a hyperactive mesocorticolimbic dopaminergic system. Intriguingly, this study found that L-theanine, a neuroprotective compound, counteracts these effects by normalising brain activity and signalling pathways, preserving cognitive and emotional functions, and preventing long-term brain dysregulation [244]. In detail, L-theanine effectively blocked Δ^9^-THC-induced cognitive and affective abnormalities, restoring normal memory functions, reducing anxiety, and preventing anhedonia. L-theanine also normalised dopaminergic signalling in both the PFC and ventral tegmental area and prevented the downregulation of the Akt/GSK-3 pathway in the PFC. Finally, L-theanine prevented the Δ^9^-THC-induced disruptions in gamma oscillations, which are essential for proper cognitive and sensorimotor gating functions. In summary, L-theanine offers hope to mitigate the detrimental effects of marijuana abuse by adolescents.

However, not only chronic CB_1_R activation can be a concern, but also long-term treatment with CBD. CBD is a negative allosteric modulator of CB_1_R, CB_2_R and GPR55, while it activates (and likely desensitises) TRPV_1_R and inhibits eCB reuptake, among other pharmacological actions [10,256,257]. The number of phytocannabinoid-based medications is steadily growing, and these formulations often contain Δ^9^-THC, CBD or both. The anticonvulsant Epidiolex is a purified CBD solution which is taken twice daily for several weeks or months by children with intractable epilepsy [24]. Even though the antiepileptic actions of CBD clearly outweigh any possible influence on brain development if administered to children and adolescents, the possible neurodevelopmental effects of CBD-containing formulas nevertheless remain a valid concern. This concern was addressed by Aguiar et al. (2023), who evaluated the consequences of long-term oral treatment of adolescent and young adult rats with CBD [227]. Treatment with a CBD-enriched cannabis extract (low Δ^9^-THC, high CBD) for 15 days did not result in any changes in body weight, locomotor activity, memory consolidation, or cognitive behaviour in healthy rats. The study showed no detrimental impact on short-term memory or locomotor behaviour, indicating the absence of adverse behavioural effects even during a sensitive period like adolescence to early adulthood (Table 1). However, the chronic treatment with the extract did induce notable changes in the glutamatergic synapses in the hippocampus. There was a reduction in the GluA1 subunit of AMPA receptors, coupled with an increase in PSD95 protein levels. That is, CBD, just like other cannabinoids, can interfere with the dynamic rearrangement and maturation of glutamatergic synapses. This, however, may contribute to neuroprotective adaptations against excitotoxicity, potentially benefiting developmentally acquired neurological disorders of excitatory synaptic transmission, such as epilepsy and autism spectrum disorder [24,258]. Additionally, the expression of GFAP (a marker of astrocytic activation) was reduced in treated animals, suggesting that the CBD-enriched extract may prevent reactive astrogliosis, which is associated with neuroinflammation and excitotoxicity. Moreover, microglial arborisation in the CA1 and CA3 hippocampal regions was reduced, indicating changes in microglial morphology, although their phagocytic activity was not significantly altered. Altogether, the study of Aguiar et al. (2023) underscores the potential safety of CBD-enriched cannabis extracts for therapeutic use in adolescents. The absence of behavioural detriments, coupled with neuroprotective changes in synaptic and glial components, suggests that such treatments may be well-tolerated, although further studies are needed, particularly regarding long-term effects [227] (Table 1).

### 3.2. The Maturating Human Brain Is Vulnerable to Cannabinoids

The human ECS undergoes significant changes during adolescence, a period marked by critical neurodevelopmental processes that affect emotional regulation, cognitive function, and vulnerability to psychiatric disorders. Emerging research suggests that the ECS is particularly sensitive to genetic polymorphisms and environmental influences, such as marijuana consumption, during this time, which can have long-term consequences on brain maturation [243,246] (Table 1). Adolescent exposure to Δ^9^-THC has been linked to persistent changes in the PFC, hippocampus and amygdala, regions critical for decision-making, memory, and impulse control. Human and rodent studies both have invariably demonstrated that Δ^9^-THC disrupts the balance of excitatory and inhibitory neurotransmission, which is essential for the refinement of synaptic connections during adolescence [255,259,260]. A recent study exploring the acute effects of cannabis on brain network connectivity has shown that cannabis disrupts multiple resting-state networks, particularly affecting the default mode, executive control, salience, hippocampal, and limbic striatal networks [228]. The authors tested the hypothesis that acute cannabis use could interfere with the undergoing significant structural changes of the PFC and hippocampus in the immature brain, thus contributing to impaired cognition and emotional processing. Using fMRI, Ertl and colleagues compared adolescents (16–17 years) and young adults (26–29 years) and found that cannabis significantly reduced within-network connectivity across these brain networks, with no significant difference between the age groups. Contrary to expectations, CBD did not attenuate the effects of Δ^9^-THC and, in some cases, exacerbated the disruptions in connectivity, further challenging the assumption that CBD can counteract the negative effects of Δ^9^-THC. These disruptions in brain network connectivity are closely tied to cognitive functions, particularly decision-making, memory, and emotional regulation, which are especially vulnerable during adolescence due to ongoing brain maturation [228] (Table 1).

As for psychiatric outcomes, cannabis use during adolescence doubles the risk of developing anxiety disorders in adulthood [259]. This risk is particularly pronounced in individuals who begin using cannabis before age 15, and it is more prevalent in females. Depressive disorders are also more common in adolescent cannabis users, and this is linked to reduced hippocampal and white matter volumes, probably because of lesser connectivity among brain regions regulating mood and emotions [261], but more direct effects on glutamate and monoamine turnovers can also be considered. Genetic variations in the ECS can also influence mental health outcomes during adolescence. Desai et al. (2024) examined how the FAAH C385A variant affects anandamide metabolism and modulates anxiety, depression, and brain activity related to threat and reward processing [230]. They found that youth with the FAAH AA genotype showed lower depressive symptoms compared to those with the AC or CC genotypes. This nonsynonymous FAAH C385A polymorphism is found in one-quarter of humans with Caucasian ancestry, and it reduces FAAH activity and thus elevates anandamide levels. The 385A allele has been associated with lower anxiety and more efficient amygdala regulation in response to stress, but also with a greater index of impulsivity, stronger reward-related activity in the ventral striatum, street drug use, problem drug/alcohol abuse, as well as obesity [231] (Table 1). The impact of FAAH polymorphism can be particularly pronounced during adolescence, when corticolimbic circuits involved in emotional regulation, such as the PFC and amygdala, are still maturing [232] (Table 1).

In addition to genetic vulnerabilities, marijuana consumption during adolescence exerts significant effects on brain development, particularly through the disruption of CB_1_R-mediated signalling. The major culprit is very likely Δ^9^-THC, the psychoactive component of drug-type cannabis preparation, which, during adolescence, has been shown to alter the trajectory of synaptic pruning and neuroplasticity in corticolimbic circuits, leading to long-term impairments in cognitive function and emotional regulation [243,259]. Clearly, early cannabis use, particularly before age 17, is linked to lasting deficits in cognitive functions such as working memory, attention, decision-making, attention, and executive functions and verbal IQ. Higher Δ^9^-THC concentrations in modern cannabis strains amplify the potential for psychiatric disorders [223,225] (Table 1). Neuroimaging studies have shown structural abnormalities, including reduced grey matter volume in the PFC, altered white matter integrity, and reduced hippocampal volume and functioning, which correlate with cognitive impairments [226,229,261]. An important and rare longitudinal study enrolling almost 800 young subjects examined how cannabis use during adolescence affects brain development, focusing on cortical thickness changes over time. Results show that greater cannabis use is associated with increased thinning in the left and right PFC 5 years after the establishment of baseline cortical thickness. However, baseline cortical thickness was not associated with experimentation with cannabis. The extent of PFC atrophy was dose-dependent and linked to attentional impulsiveness at follow-up [262].

Notwithstanding, it is still largely debated to which extent adolescent marijuana use affects brain development. A recent systematic review and meta-analysis of voxel-based morphometry studies investigated the overall effects of adolescent cannabis use on brain morphology, with a focus on age, sex, and grey matter volume (GMV) differences [263]. Curiously, when combining all six included studies, no significant GMV differences were found between cannabis-using youth and typically developing youth. The study identified age-related GMV changes in the left superior temporal gyrus (L-STG). The L-STG is involved in auditory, speech, language, and emotional processing. Structural abnormalities in this region could contribute to impairments in social cognition and increase the risk of psychotic or affective disorders, particularly since cannabis use is associated with higher risks for these conditions in adolescence. Supplemental analyses found that a longer duration of cannabis use was associated with decreased GMV in the L-STG, supporting the idea that cumulative cannabis exposure may contribute to structural brain changes [263]. Older cannabis user youth showed decreased GMV compared to age-matched cannabis-naïve youth, while younger cannabis user youth showed increased GMV. This suggests a developmental gradient, with cannabis exposure potentially affecting GMV differently, depending on the age at which cannabis use occurs. A meta-regression revealed that studies with a higher proportion of female participants showed increased GMV in the right middle occipital gyrus in cannabis-user youth compared to typically developing youth. Conversely, in studies with a higher proportion of males, cannabis-user youth showed decreased GMV in this region. This indicates that sex may moderate the relationship between cannabis use and brain morphology, with females showing different neuroanatomical effects of cannabis compared to males. These differences may be accounted for by hormonal influences or differences in cannabis-related behaviour between the sexes. These findings can be best explained assuming that cannabis-related GMV increases in younger adolescents may be due to disrupted synaptic pruning, while in older adolescents or young adults, a reduced GMV may be a result of accelerated pruning and neurotoxic processes [263]. As discussed, significant sex differences may also exist, with female [150] and male [151] rats displaying distinct profiles in synaptic pruning impairment. Notably, cannabinoids may indirectly counteract their direct effects on synaptic activity-dependent pruning by impairing the role of microglia in this critical process.

A similar pattern emerges from another study examining the impact of early (under 16) versus late (16 and older) onset of marijuana use on cortical brain structure, focusing on three key measures: cortical thickness, grey–white matter contrast (GWR), and local gyrification index (LGI). In particular, it explores how continued marijuana use after adolescence might lead to lasting changes in brain development [264]. The authors found that early-onset users showed an association between continued marijuana use and increased cortical thickness, greater GWR, and reduced LGI. This was particularly noted in the anterior dorsolateral prefrontal cortex, a region critical for cognitive functions. Late-onset users, in contrast, exhibited thinner cortex and reduced GWR with increased marijuana use, showing an opposite trajectory. Notably, these findings are remarkably similar to those seen in the late-adolescent rats [150,151], that is, an accelerated pruning-induced cortical atrophy, changes that are not seen with later exposure.

These studies altogether suggest that early-adolescent marijuana exposure disrupts typical adolescent pruning processes, which involve synaptic elimination and refinement to streamline neural connections. The observed increase in cortical thickness and GWR in early-onset users could indicate that pruning is less efficient (perhaps at the grey–white boundary), possibly leading to a thicker, less refined cortex as fewer synaptic connections are eliminated. Nevertheless, the question always lingers over these studies: are the findings meaningful? This study of Filbey et al. [264] also has its limitations. The study is cross-sectional, which limits causal conclusions. Additionally, behavioural or cognitive data were not correlated with these morphological changes, which would provide insight into functional impacts. Nicotine use and potential breaks in marijuana use history could also influence the results.

All in all, these studies emphasise that more longitudinal research is needed to disentangle the complex relationships between age, sex, cannabis exposure, and brain development, especially during the critical period of adolescence, and additional confounding factors also need to be considered, including alcohol and tobacco use, socioeconomic status, the strength of marijuana strains consumed and the mode of ingestion. Another longitudinal study assessed the cognitive performance of over 1000 individuals born after 1972 [226]. Initial neuropsychological testing was conducted at age 13 before any cannabis use had begun. The participants had varying histories of cannabis use, ranging from non-use to cannabis dependence. Follow-up assessments were completed when the participants reached age 38. Persistent cannabis users exhibited significant impairments in memory function, including challenges with both short-term memory (working memory) and long-term memory retention. This decline was observed across multiple domains of neuropsychological testing and was particularly severe in individuals who started using cannabis in adolescence (Table 1).

Cannabis users, especially those who started young, also showed marked deficits in executive functioning, such as problem-solving, decision-making, planning, and the ability to inhibit impulsive behaviour. One of the notable declines was in processing speed, the cognitive ability to quickly and efficiently perform mental tasks. Slow processing speed can make it difficult for individuals to follow instructions, keep up with conversations, or respond quickly in demanding environments. Persistent cannabis users, particularly those with adolescent-onset use, showed slower processing speeds over time. Cannabis users also experienced significant problems with sustained attention and focus. This manifested as distractibility, difficulty concentrating for long periods, and an inability to stay engaged with tasks. These issues were noticeable not just in test results but also in daily life, as reported by friends and family members of the participants. Finally, the study found a clear association between persistent cannabis use and a measurable decline in IQ. Those with the most severe decline in IQ were individuals who started using cannabis during adolescence and continued using it persistently. This study by Meier et al. (2012) thus clearly confirms that cannabis use during brain development may have a neurotoxic effect, leading to long-lasting cognitive impairments, with IQ drops as significant as 6 to 8 points over the span of the study [226].

Marijuana use, particularly during adolescence, is also strongly associated with an increased risk of psychosis. It is easy to understand why since the ECS controls the development of all domains and systems which are affected in schizophrenia, including certain brain areas (PFC, hippocampus, amygdala, striatum, L-STG), GABAergic and glutamatergic signalling, monoaminergic neuromodulation and even brain metabolism [224]. This risk can manifest as temporary psychotic episodes or symptoms, but in some cases, it may persist and contribute to the development of more chronic conditions, such as schizophrenia. While marijuana use during adolescence may elevate the risk of schizophrenia in some individuals, the association between marijuana and schizophrenia is more complex and less direct than its link to psychosis. Schizophrenia is a chronic mental disorder that typically emerges in late adolescence or early adulthood, characterised not only by psychotic symptoms but also by cognitive impairments and negative symptoms like social withdrawal [223,224,260]. Longitudinal studies indicate that adolescent marijuana users, especially those who use it frequently or consume high-potency strains, are at a higher risk of developing schizophrenia later in life. However, marijuana use alone is unlikely to cause schizophrenia; rather, it may act as a trigger in individuals who are genetically predisposed, e.g., those carrying variants in their catecholamine-O-methyltransferase (COMT) gene or in their CB_1_R gene CNR1. This is supported by the observation that while adolescent marijuana use is a growing problem, the incidence of new schizophrenia cases has not shown a corresponding increase. Additionally, it is possible that individuals with a genetic predisposition to schizophrenia are more likely to experiment with marijuana during adolescence, further complicating the relationship between marijuana use and schizophrenia risk [224].

The following ground-breaking study by Tao et al. (2020) shed new light on how genetic predispositions, environmental influences, and marijuana use converge in the development of schizophrenia [212]. They found that in the PFC and the hippocampus, CB_1_R mRNA expression is highest in the foetal period, followed by a sharp decline postnatally, which stabilises throughout adulthood. This strongly implies that CB_1_R activity is critical during human brain development. Notably, carriers of the COMT Val158 allele showed a stronger negative correlation between CNR1 expression in the dorsolateral PFC (DLPFC) and age, potentially linking cannabis exposure during adolescence to dysregulated brain development. Furthermore, CNR1 expression was significantly decreased in the DLPFC of patients with schizophrenia and major depressive disorder, suggesting that ECS dysregulation is involved in the pathology of these psychiatric conditions. Interestingly, Δ^9^-THC or ethanol exposure upregulated CNR1 expression in patients with affective disorders, and CNR1 expression was also increased in schizophrenia patients who completed suicide, pointing to the complex interaction between cannabis use, mental health, and suicide risk. DNA methylation at specific loci (e.g., cg02498983) correlated with age and COMT genotype in the PFC. Carriers of the Val158 allele showed the steepest increase in methylation over time, and this negatively correlated with CNR1 expression. This correlates well with the above animal studies, suggesting that epigenetic modulation induced by environmental factors, including marijuana abuse, can reprogram brain circuits during adolescence, increasing the risk of psychosis. Additionally, the study identified a novel CNR1 transcript, whose expression was associated with a single nucleotide polymorphism rs806368, a genetic variant previously linked to substance dependence. This transcript might regulate CB_1_R expression in response to cannabis exposure, contributing to the development of addiction and psychiatric disorders in genetically predisposed individuals [212].

Although the level of expression (mRNA) and protein density are not interchangeable terms, most studies reported in this review agree that both peak at early stages of brain development. We reported a steady decline in rat hippocampal CB_1_R density during the postnatal life [237]. However, we also found much higher CB_1_R density in the embryonic hippocampus, with a steep decline until birth (unpublished). A post-mortem study also found that CB_1_R mRNA expression in the human DLPFC decreases significantly over time, peaking during neonatal life and declining steadily into adulthood [236] (Table 1). This pattern was particularly evident in cortical layer 2, suggesting that eCB-mediated regulation of neurotransmission is robust in early life but diminishes with age. DAGLα expression followed a bell-shaped curve, with low levels in infancy and adulthood but peaking during school age to young adulthood. This suggests that the production of 2-AG is particularly important during cognitive development in childhood. While the typically presynaptic expression of MAGL declined after infancy, the expression of the postsynaptic 2-AG-metabolizing enzyme, ABHD6, showed a steady increase across development. This may reflect a developmental switch from retrograde inhibition to dendritic self-inhibition [16]. In contrast, both NAPE-PLD and FAAH steadily increased from infancy to adulthood, indicating that AEA becomes increasingly important after adolescence. CB_1_R mRNA was highly expressed in cortical layer 2 during early life (neonates and toddlers), while the deep cortical layers 5 and 6 showed weaker but still significant CB_1_R mRNA expression. CB_1_R expression decreased significantly with age, particularly in superficial layers like 2 and 3, and the intensity of expression in the deeper layers (5 and 6) also declined by adulthood. Notably, CB_1_R mRNA showed a clear association with GABAergic interneuron markers, supporting the notion about the role of CB_1_R in early-life regulation of cortical interneuron development [236] (Table 1).

Additional post-mortem studies in patients with schizophrenia reveal a strong GABAergic dysfunction in the corticolimbic areas, particularly of the parvalbumin^+^ GABAergic neurons, leading to impaired inhibitory control of pyramidal neurons and disrupted gamma oscillations, which are essential for cognitive processing, together with the hyperactivity of the mesolimbic dopaminergic system [223,259,265]. The negative symptoms (alogia, anhedonia, affective flattening, avolition, memory problems, and social withdrawal) are mostly linked with hypofrontality and, more closely, disturbances in GABAergic and glutamatergic activities of the PFC. The positive symptoms of schizophrenia (hallucinations, paranoia, disorganised thinking, abnormal motor behaviour) are closely linked with a hyperdopaminergic state, particularly in the mesolimbic pathway [264]. As the animal studies made very clear, chronic exposure to CB_1_R agonists during adolescence indeed causes hypofrontality and hyperdopaminergic state via multiple mechanisms, consistent with lasting developmental, neurochemical and neurophysiological changes in the corticolimbic system and beyond [260]. While acutely, Δ^9^-THC administration in humans induces several schizophrenia-like symptoms, including paranoia, hallucinations and cognitive impairments, in the long run, Δ^9^-THC exposure can exacerbate psychotic symptoms in individuals already diagnosed with schizophrenia, or it can facilitate the onset of schizophrenia in individuals with genetic predisposition [223,224,266]. These effects have a strong neurodevelopmental component when marijuana abuse occurs during adolescence [260].

Importantly, CBD has been proposed as a possible antipsychotic medicine [260,267], with proven therapeutic potential against a multitude of complications in schizophrenia, including the following:Positive Symptoms: CBD has been shown to ameliorate hyperlocomotion and stereotypies, which are proxies for positive symptoms like psychomotor agitation and hallucinations in schizophrenia. CBD may exert antipsychotic effects by normalising dopamine signalling and counteracting Δ^9^-THC’s psychotomimetic effects;Negative Symptoms: There is evidence that CBD can improve social interaction deficits and reduce immobility in animal models of schizophrenia, suggesting it could treat negative symptoms such as social withdrawal, anhedonia, and lack of motivation;Cognitive Symptoms: CBD has shown promise in reversing cognitive deficits in preclinical models, particularly in memory and attention tasks. It has been shown to restore object recognition memory and working memory, likely by modulating PFC and hippocampal circuits.

CBD’s antipsychotic effects may stem from its ability to modulate CB_1_Rs, CB_2_Rs and TRPV_1_Rs by affecting AEA turnover, by acting as a partial D_2_R/D_3_R agonist and as a partial 5-HT1AR agonist, thus normalising monoaminergic signalling and conferring antidepressant and antipsychotic effects [260,267]. Nevertheless, an increasing body of studies has failed to provide direct evidence that CBD can counteract the psychotomimetic effects of Δ^9^-THC [228,268], thus adding to the complexity of the role of cannabis preparations in psychosis.

## 4. Conclusions

The intricate role of the ECS in brain development has been well-documented, with its influence beginning as early as embryogenesis and continuing through key developmental stages, including adolescence. Despite extensive research highlighting both its regulatory functions and its vulnerabilities, significant gaps remain in our understanding. For instance, the impact of cannabis exposure, both in utero and during adolescence, presents a multifaceted challenge. While studies consistently show that early exposure to Δ^9^-THC can disrupt brain development and maturation, particularly in the PFC and hippocampus, the exact mechanisms remain incompletely understood. While much of the focus has been on the immediate effects of other substances, such as alcohol and tobacco, the long-term implications of prenatal cannabis exposure should not be overlooked. As cannabis becomes increasingly legalised and socially accepted in many regions, the potential for underestimating its risks to foetal development grows. This calls for heightened awareness, education and caution among healthcare providers and the general public, ensuring that expectant mothers are fully informed about the potential consequences of cannabis use during pregnancy.

One of the key takeaways from this review is the need for more longitudinal studies that track the effects of cannabis exposure across the lifespan. On the one hand, there are undeniable discrepancies in findings related to cannabis-induced neurodevelopmental damage. While animal models provide robust evidence of Δ^9^-THC’s negative impact on synaptic pruning, memory, and emotional regulation, human studies yield mixed results, particularly regarding the role of genetic predispositions. Studies like those investigating FAAH and CNR1 polymorphisms suggest that genetic vulnerabilities may modulate the effects of cannabis, underscoring the importance of personalised approaches in future research and potential interventions. On the other hand, existing research provides compelling evidence that adolescence is a critical window for ECS modulation with possible or putative long-term consequences, yet most studies focus on short-term outcomes. Longitudinal research is necessary to determine whether cannabis-induced changes observed in the adolescent brain persist into adulthood and how they manifest in long-term cognitive, emotional, and psychiatric health outcomes. One of the central questions in our review is how the detrimental effects of Δ^9^-THC or synthetic cannabinoids are masked by various confounding factors. These factors include timing (early vs. late adolescence), sex differences, and genetic predispositions that might differentially impact synaptic activity-driven pruning versus microglia-mediated synaptic remodelling. Additionally, it remains uncertain whether chronic cannabinoid agonist exposure drives these changes through frequent CB_1_R overactivation or, conversely, through CB_1_R desensitisation.

The role of CBD, often proposed as a counterbalance to Δ^9^-THC’s detrimental effects, remains controversial, with some studies suggesting it may exacerbate rather than mitigate the disruptions caused by Δ^9^-THC. Thus, the field needs to address the ongoing debate about CBD’s protective or harmful effects in the context of adolescent brain development. Although CBD is touted for its neuroprotective properties and potential therapeutic applications, conflicting data point to the need for caution in using CBD-based therapies in adolescents. More detailed mechanistic studies are needed to clarify how CBD interacts with the ECS during this vulnerable period and whether it truly mitigates the risks posed by Δ^9^-THC or other cannabinoids.

In conclusion, while significant progress has been made in understanding the ECS’s role in brain development and its disruption by exogenous cannabinoids or by ECS polymorphism, more comprehensive research is needed. Specifically, longitudinal human studies, attention to genetic variability, and careful examination of therapeutic cannabinoids like CBD are crucial for filling the current knowledge gaps. Only through such efforts can we fully appreciate the complex relationship between cannabis and neurodevelopment, ensuring that both public health policies and clinical practices are informed by the latest, most reliable data.

## Figures and Tables

**Figure 1 cells-13-01875-f001:**
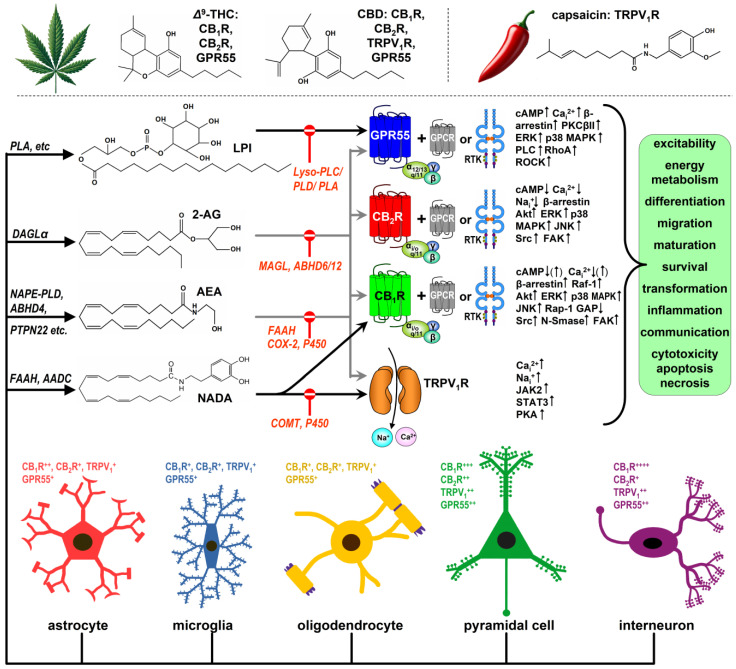
Overview of the endocannabinoid system in the brain. The endocannabinoid system (ECS) was uncovered through research investigating the molecular targets of key phytocannabinoids found in Cannabis sativa, particularly Δ^9^-tetrahydrocannabinol (Δ^9^-THC), the psychoactive component, and cannabidiol (CBD), a non-psychoactive compound. Both Δ^9^-THC and CBD interact with numerous targets within the brain, and here we focus on four key receptors: the cannabinoid receptors CB_1_ and CB_2_ (CB_1_R and CB_2_R), GPR55, and the transient receptor potential vanilloid type 1 (TRPV_1_) receptor. CB_1_R, CB_2_R, and GPR55 are G protein-coupled receptors (GPCRs) with seven transmembrane-spanning domains. These four receptors are expressed across various brain cell types, including astrocytes, microglia, oligodendrocytes [22], glutamatergic neurons, GABAergic interneurons, and projection neurons (GABAergic, monoaminergic, and cholinergic), at highly variable densities, depending on cell types and factors like brain region, age, and neuropsychiatric conditions. For instance, CB_1_Rs are present at high levels in CCK^+^ cortical and hippocampal GABAergic interneurons and at moderate levels in VGLUT1^+^ pyramidal cells but are virtually absent in parvalbumin^+^ interneurons and VGLUT2^+^ pyramidal cells. Although synaptic pruning is assisted by resting microglia, which expresses low levels of CB_1_Rs and CB_2_Rs, when activated, these cells express much higher amounts of both receptors. All four receptors are typically found in the cytoplasmic membrane—primarily in nerve terminals, dendrites, and cell bodies— albeit there is substantial evidence for their intracellular localisation, too. While Δ^9^-THC acts as a partial agonist at these GPCRs, CBD’s pharmacological actions are more complex, often resembling negative allosteric modulation at the CB_1_R and the CB_2_R and weak partial agonism at the CB_1_R-CB_2_R heterodimer [10,23]. CBD is a functionally selective antagonist at the GPR55 because it inhibits agonist-induced G-protein signalling, while it does not affect β-arrestin-mediated pathways [24]. Whereas CBD at pharmacologically high concentrations (≥10 µM) can activate and desensitise the ionotropic TRPV_1_R, at therapeutically relevant concentrations (≤1 µM), CBD only deactivates the TRPV_1_R [25]. In addition to receptors, the ECS includes enzymes responsible for synthesising lipid ligands that activate these receptors. One of the most well-studied eCBs, anandamide (N-arachidonoyl-ethanolamine or AEA), is synthesised from N-acylphosphatidylethanolamine (NAPE) via NAPE-specific phospholipase D (PLD). Several alternative pathways also contribute to AEA production. Diacylglycerol lipase α (DAGLα) is the primary enzyme that synthesises 2-arachidonoyl-glycerol (2-AG), another major eCB. Both AEA and 2-AG activate all four receptors, though other ligands exhibit more receptor-selective actions. For example, N-arachidonoyl-dopamine (NADA), likely produced by fatty acid amide hydrolase (FAAH) in dopaminergic cells, acts as a hybrid agonist for CB_1_R and TRPV_1_R [26]. Similarly, L-α-lysophosphatidyl-inositol (LPI) and its congeners resemble classical eCBs but selectively activate GPR55. The activation of these receptors can influence virtually all functions of the brain cells expressing them, but their actions are highly context-dependent. The effects depend on factors such as receptor splice variants, heteromeric interactions with other receptors (e.g., TrkB, insulin receptor, or EGF receptor), the cell’s metabolic state and age, and the ontogenetic stage of the organism. For instance, homo(di)meric CB_1_Rs in neurons are mostly coupled to G_i/o_ proteins and inhibit cAMP production, but in astrocytes, CB_1_R activation stimulates G_q_ and, consequently, [Ca^2+^]_i_ levels. Heteromeric CB_1_Rs, such as the adenosine A_2A_ receptor-CB_1_R heterotetramer, may also couple to G_s_ and stimulate cAMP synthesis [10]. Many receptor-mediated effects are tied to brain cell processes, such as differentiation, maturation, migration, circuit formation, and plasticity, which are key topics in this review. Finally, after eCBs activate their receptors, they are primarily metabolised intracellularly by a variety of enzymes. The key enzymes for this review are FAAH and cyclooxygenase-2 (COX-2), which degrade anandamide, and monoacylglycerol lipase (MAGL), which metabolises 2-AG. Cytochrome P450 (P450) enzymes may also contribute to eCB metabolism. LPI is broken down by various lysophospholipases (A, C, and D).

**Figure 2 cells-13-01875-f002:**
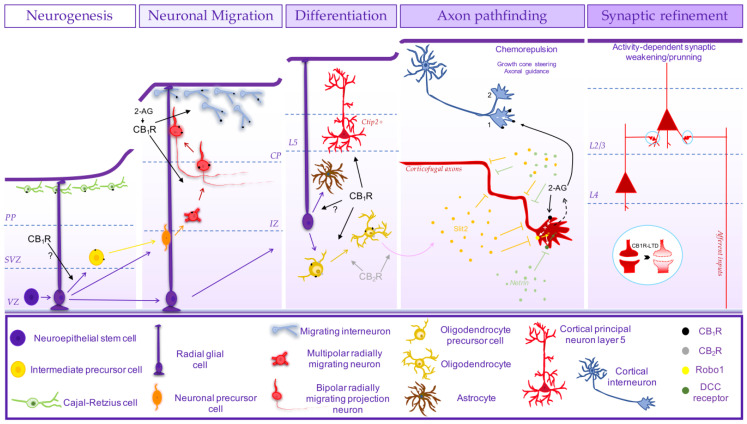
Schematic representation of the involvement of endocannabinoid signalling system (ECS) in corticogenesis. ECS through CB_1_R may be involved in cortical cell proliferation [92,94,133] and intermediate precursor cell generation [101]. CB_1_R is expressed in Cajal-Retzius cells and may control early-born cortical projection neuron positioning [93,100,104]. CB_1_R is involved in radial migration [92] in the transition from the intermediate zone (IZ) towards the cortical plate (CP) [106] by controlling the neuronal polarisation [102] and eventually through the control of cell movement [102,139], controlling the distribution of neurons across the different cortical layers [92,106]. CB_1_R also regulates tangential migration of cortical interneurons [37,104]. ECS, via CB_1_R, is also engaged in the development of cortical excitatory cytoarchitecture by controlling the differentiation of cortical projection neurons of layer 5 (Ctip2^+^) [94,109,128]. Next, ECS controls the guidance of corticofugal axons [92,94,103,110,124] by regulating growth cone steering through autocrine signalling by 2-AG via CB_1_R at the growth cones, through the regulation of Robo1 receptor and the concomitant CB_2_R-induced release of Slit2 by oligodendrocytes [124], whose differentiation was shown to entail CB_1_R and CB_2_R [124,140,141,142,143]. ECS may also control axon pathfinding through the regulation of the trafficking of deleted in colorectal cancer (DCC) receptors, which tethers the action of the guidance cue netrin [144]. 2-AG signalling through CB_1_R also controls growth cone steering of cortical interneurons via RhoA activation [108]. ECS is later involved in cortical synaptic refinement by controlling synaptic weakening/pruning through CB_1_R-mediated long-term depression (LTD), observed in afferent inputs at layer 2/3 and layer 4-layer 2/3 synapses [145,146,147,148,149,150,151]. VZ, Ventricular Zone; SVZ, Subventricular zone; PP, Preplate.

**Table 1 cells-13-01875-t001:** The involvement of the endocannabinoid system in adolescent brain development.

Receptor Enzyme Ligand	Function in the Adolescent Brain	Effects of External Cannabinoids	Consequences if Perturbed	Sex-Dependent Effects	References
**CB_1_R**	Regulates excitatory/inhibitory neurotransmission, synaptic pruning, and maturation of corticolimbic circuits (e.g., PFC, hippocampus). Peaks during adolescence, declines in adulthood.	Δ^9^-THC acts as a CB_1_R agonist. Chronic exposure downregulates CB_1_R, desensitises receptors, impairs synaptic plasticity, and reduces dendritic complexity.	Persistent changes in PFC and hippocampal structure. Increased risk of psychiatric disorders (e.g., anxiety, schizophrenia). Impaired executive function, memory, and emotional regulation.	Greater CB_1_R density in males, more efficient CB_1_R coupling in females. More pronounced cognitive and emotional impairments in female rodents. Males show delayed onset of CB_1_R-mediated synaptic plasticity.	[215,216,217]
**CB_2_R**	Involved in immune regulation and neuroinflammation. Low neuronal expression in the brain but increases in microglia with neuroinflammation.	Chronic Δ^9^-THC exposure reduces CB_2_R density in adolescent brains. Selective acute CB_2_R activation (e.g., by AM1241) can reduce neuroinflammation and prevent anxiety-like behaviours during adolescence.	Chronic Δ^9^-THC exposure unequivocally downregulates CB_2_R expression, which may exacerbate anxiety and neuroinflammation caused by substance abuse or stress.	Two-fold greater CB_2_R expression in adolescent but not adult females.	[216,218,219]
**TRPV_1_R**	Involved in modulating stress and anxiety responses during adolescence. Opposes CB_1_R effects on anxiety regulation.	TRPV_1_R activation by CBD or stress can exacerbate anxiety responses. TRPV_1_R-dependent LTP in the hippocampus may be linked to cognitive deficits caused by alcohol exposure.	Increased anxiety and cognitive deficits when activated by cannabinoids or stress. TRPV_1_ blockade may provide therapeutic potential for treating anxiety disorders.	Females show earlier onset of TRPV_1_-mediated synaptic plasticity. Male rodents show stronger anxiety-related responses to TRPV_1_ activation.	[216,220,221]
**Δ^9^-THC**	Partial agonist of the CB_1_R and the CB_2_R. Interferes with the maturation of corticolimbic circuits, synaptic pruning, and neuroplasticity during adolescence.	May lead to downregulation and desensitisation of its receptors with chronic use. Disrupts synaptic plasticity, reduces dendritic complexity, and impairs signalling in the PFC and hippocampus. Triggers hypoGABAergic and hyperdopaminergic state.	Affects cortical thickness and wiring. Long-lasting cognitive impairments (e.g., memory, decision-making) and emotional dysregulation. Increases the risk of psychiatric disorders like anxiety, depression, and schizophrenia.	Females are more susceptible to Δ^9^-THC-induced emotional and cognitive impairments, showing greater downregulation of CB_1_R. Males tend to exhibit delayed onset of Δ^9^-THC-induced synaptic plasticity changes.	[217,222,223,224,225,226]
**CBD**	Negative allosteric modulator of CB_1_R and CB_2_R. Activates TRPV_1_R and inhibits eCB reuptake. Potential neuroprotective role during brain development.	Long-term CBD exposure can affect glutamatergic synapses and synaptic plasticity. May have neuroprotective effects but can also exacerbate disruptions in brain network connectivity when co-administered with Δ^9^-THC.	Reduction in GluA1 AMPA subunit and increased PSD95. Alters brain connectivity, especially when combined with Δ^9^-THC. No adverse effects on cognitive or motor functions in healthy adolescents.	Males may experience greater cognitive protection from CBD. Females show increased susceptibility to CBD’s effects on synaptic plasticity when combined with Δ^9^-THC.	[227,228,229]
**FAAH**	Breaks down anandamide. Controls anandamide levels and regulates emotional responses, stress, and cognitive functions.	Polymorphism FAAH C385A reduces enzyme activity, leading to elevated anandamide levels and altered stress responses. Chronic Δ^9^-THC exposure interferes with FAAH activity, but reports are conflicting.	Reduced FAAH activity is associated with heightened emotional regulation and impulsivity but can increase susceptibility to substance abuse and psychiatric disorders.	Females generally have lower FAAH expression during adolescence, leading to prolonged anandamide signalling. Males with FAAH C385A polymorphism show stronger reward-related activity, impulsivity and risk-taking behaviours.	[216,218,230,231,232,233,234,235]
**MAGL**	Breaks down 2-AG. Regulates synaptic plasticity, excitatory/inhibitory balance, and emotional regulation.	Decrease in function during adolescence. Chronic Δ^9^-THC exposure reduces microglial but increases overall MAGL expression, leading to altered synaptic transmission.	Dysregulation of synaptic connections and plasticity in corticolimbic circuits. Long-term emotional and cognitive deficits.	Inhibiting MAGL uncover LTD in juvenile males.	[216,218,235,236]
**DAGLα**	Synthesises 2-AG, essential for synaptic plasticity and connectivity in brain maturation. Peaks during adolescence.	Its expression peaks during adolescence. Δ^9^-THC can alter DAGL activity, affecting 2-AG synthesis and overall cannabinoid signalling during brain development.	Disruption in the production of 2-AG leading to impaired synaptic connectivity, memory, and emotional regulation.	Females show earlier DAGL maturation and heightened synaptic plasticity, whereas males exhibit more delayed effects on synaptic development.	[210,235,237]
**NAPE-PLD**	Synthesises anandamide, plays a role in regulating emotional and cognitive functions.	Gain of function during adolescence. Chronic Δ^9^-THC exposure can reduce NAPE-PLD expression in microglia, leading to altered anandamide production.	Impaired emotional regulation and stress response. Increased risk of psychiatric disorders due to disrupted anandamide signalling.	Females show higher baseline NAPE-PLD activity, contributing to sex differences in emotional regulation under stress or cannabinoid exposure.	[216,218,236]
**ABHD6**	Degrades 2-AG, plays a role in regulating synaptic plasticity and emotional responses.	Chronic Δ^9^-THC exposure can increase ABHD6 expression in the placenta but not in the brain. ABHD6 expression is increased in the PFC of schizophrenic adolescents.	Dysregulated synaptic transmission and plasticity in corticolimbic circuits leading to cognitive and emotional deficits.	Males show higher ABHD6 levels during adolescence, leading to stronger inhibition of 2-AG signalling compared to females.	[216,238,239]

## Data Availability

No new data were created or analysed in this study.

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
