# Peer review of "Cannabis, Endocannabinoids and Brain Development: From Embryogenesis to Adolescence"

_cells, 2024, doi:10.3390/cells13221875_

Round 1
Reviewer 1 Report
Comments and Suggestions for Authors
This is an excellent review, useful not only for specialists in the field of cannabinoids, but also for medical doctors working with patients with cannabis-related problems. The authors included in their manuscript effects of drug abuse (alcohol) and stress on the endocannabinoid system, which further increased the value and importance of this review.
Remark
It is worth to explain in the beginning of the article (and not only in the conclusion) that increased use of cannabis in all age groups and thus also among pregnant women make this review especially actual and important.
Suggestions
Chapter 1.2 a; Mention the huge difference in CB1 expression between neuronal types
b; Microglia express CB1 only after activation and at a very low level (order of magnitude less than CB2).
Line 100 (and also Table 2 at page 25) Please add that GPR55 is probably the main receptor of CBD (at least its anti-epileptic effect is dependent on GPR55 expression).
Figure 1 I really liked this complex figure, I think it will be widely used for lectures about the cannabinoid system. Nevertheless, as the Authors also wrote, the typical change in cAMP level is a decline after CB1 activation, so please correct the figure accordingly. The presence of CB1 in microglia is questionable. Here the authors could somehow indicate high or low expression at the cell types. Indeed, by all cell -types the same four receptors are always written only with different colours. In this form, it is not very informative.
Line 151 Add that CBD activates GPR55.
Line 227 It seems to be trivial but could be mentioned that from the appearance of CB1 expression the developing embryonic brain is potentially sensitive to THC.
Lines 610-612 – It is worth to add that 2-AG is the major ligand of CB1, with 2-3 orders of magnitude higher concentration.
Minor points
Line 4 What indicates the number 3 after the author names?
Line 38-40 Delete “drug-naïve” – otherwise it is easy to misunderstand the sentence
Line 72 Delete “homodimer” – in their more usual monomeric form they also couple predominantly with Gi proteins.
Line 77 Add “depending on cell type and area” because activation of CB1 does not lead activation of all these signaling pathways generally.
Lines 199-202 Delete “earliest stage of nervous system development” it is written twice in the same sentence.
Lines 524-529. -I think these sentences are misplaced in this chapter, they should appear at chapter 3.1 and 3.2
Figure 2 – Again an excellent illustration. I suggest the change the colour of the oligodendrocyte, it is practically identical in the submitted version as the background.
Lines 823-824 – Please change wording, because if the negative effect still remains (as written in line 823) than the study of Aguiar did not really allay it.
Table 2 by FAAH D9 appears twice

Author Response
Comments 1: This is an excellent review, useful not only for specialists in the field of cannabinoids, but also for medical doctors working with patients with cannabis-related problems. The authors included in their manuscript effects of drug abuse (alcohol) and stress on the endocannabinoid system, which further increased the value and importance of this review.
Response 1: We thank very much the Reviewer for the thoughtful and positive review, and for the comments that contributed to improve the quality of the manuscript. The modifications introduced to the revised manuscript are indicated using track changes.
Remark
Comments 2:
It is worth to explain in the beginning of the article (and not only in the conclusion) that increased use of cannabis in all age groups and thus also among pregnant women make this review especially actual and important.
Response 2: Thank you for this valuable suggestion. We decided to restructure the beginning of the first chapter, as we had not included information on the epidemiology of cannabis use. This change allowed us to incorporate recent studies on post-legalization usage statistics of cannabis during different stages of pregnancy and adolescence. We believe that the new Section 1.1 and the Conclusions provide a cohesive framework for our review.
Suggestions
Comments 3: Chapter 1.2 a; Mention the huge difference in CB1 expression between neuronal types
Response 3: We amended the text accordingly. See the full sentence in Response 4. We also address this issue by introducing modifications in Figure 1 and its legend (see Response 6).
Comments 4: b; Microglia express CB1 only after activation and at a very low level (order of magnitude less than CB2).
Response 4: The question of cannabinoid receptor levels, particularly CB1R, in various brain cells remains unresolved, as no study provides a side-by-side comparison of the expression and density of different cannabinoid receptors across all major brain cell types. Receptor expression in any cell type – as one can infer it from our review – also depends on the developmental stage. Some, including Dr. Nephi Stella, wonder if CB1Rs and CB2Rs are present at all in resting microglia (Stella N. Cannabinoid and cannabinoid-like receptors in microglia, astrocytes, and astrocytomas. Glia. 2010 Jul;58(9):1017-30. https://doi.org/10.1002/glia.20983.). Nevertheless, microglia activation can bring their expression up, thus technically, these receptors take part in some microglial life cycle.
But there is another issue. It is well-known that resting microglia contribute to synaptic pruning during adolescence (Cornell et al., Microglia regulation of synaptic plasticity and learning and memory, Neural Regen Res. 2022 Apr;17(4):705-716. https://doi.org/10.4103/1673-5374.322423) and we highlighted a study by Daniele Piomelli’s group (Lee et al., Biol. Psychiatry 2022, 92(11), 845-860. https://doi.org/10.1016/j.biopsych.2022.04.017), which concluded that frequent low-dose Δ9-THC exposure during adolescence disrupts microglial homeostasis and impairs responses to microbial infection and social stress in young adulthood.
The Reviewer's suggestion led us to investigate further, revealing strong evidence that CB1R in resting microglia is a significant factor in some of Δ9-THC's adverse effects on social memory (Hasegawa et al., Microglial cannabinoid receptor type 1 mediates social memory deficits in mice produced by adolescent THC exposure and 16p11.2 duplication, Nat. Commun. 2023, 14(1), 6559. https://doi.org/10.1038/s41467-023-42276-5). Notably, the study demonstrated that microglia-specific deletion of CB1R restored normal phenotypes in adolescent rats treated with Δ9-THC. Consequently, we incorporated this study of Hasegawa et al. (2023) in Chapter 3, and also amended the text on CB1R expression across different cell types in Chapter 1, as follows (page 3 lines 103-108): “Although no study has systematically compared CB1R and CB2R densities across brain cell types, it is widely accepted that hippocampal and neocortical GABAergic interneurons have the highest CB1R density in the brain, while pyramidal neurons have much lower CB1R levels. Probably even lower levels are found in other neuron types and astrocytes. Marginal CB1R expression is expected in microglia, oligodendrocytes, oligodendrocyte precursor cells, and adult neural stem cells (Lutz, 2020; Hasegawa et al. 2023).”
We also addressed this issue by introducing modifications in Figure 1 and its legend (see Response 6).
Comment 5: Line 100 (and also Table 2 at page 25) Please add that GPR55 is probably the main receptor of CBD (at least its anti-epileptic effect is dependent on GPR55 expression).
Response 5: We are highly interested in the complex involvement of the eCB system in epileptogenesis. However, two contrasting views currently exist on the mechanisms by which CBD exerts its antiepileptic effects. Even among those who believe that CBD’s antiepileptic action is GPR55-dependent, opinions are divided: one group argues that GPR55 functions as an excitatory receptor in glutamatergic pre-terminals, where it enhances circuit hyperexcitability. Consequently, GPR55 blockade reduces excitation and raises the epilepsy threshold (Sylantyev et al., 'Cannabinoid- and lysophosphatidylinositol-sensitive receptor GPR55 boosts neurotransmitter release at central synapses,' Proc Natl Acad Sci U S A. 2013, 110(13):5193-5198. https://doi.org/10.1073/pnas.1211204110). Dr. Ruth Ross stands out as an internationally renowned investigator in this group.
Conversely, another group, as reported in a separate PNAS study, concluded that GPR55 acts as an inhibitory receptor in GABAergic cells. Thus, GPR55 blockade by CBD increases GABAergic neurotransmission and raises the epilepsy threshold (Kaplan et al., 'Cannabidiol attenuates seizures and social deficits in a mouse model of Dravet syndrome,' Proc Natl Acad Sci U S A. 2017, 114(42):11229-11234. https://doi.org/10.1073/pnas.1711351114). This research is associated with Dr. Nephi Stella’s group.
A third group found clear evidence for the involvement of TRPV1R, which CBD desensitizes (Gray et al., 'Anticonvulsive Properties of Cannabidiol in a Model of Generalized Seizure Are Transient Receptor Potential Vanilloid 1 Dependent,' Cannabis Cannabinoid Res. 2020, 5(2):145-149. https://doi.org/10.1089/can.2019.0028). Dr. Vincenzo Di Marzo is a key figure in this research.
Given that respected cannabinoid researchers still hold differing perspectives on this topic, we have chosen to remain neutral, particularly as epilepsy is beyond the scope of our review.
Comments 6: Figure 1 I really liked this complex figure, I think it will be widely used for lectures about the cannabinoid system. Nevertheless, as the Authors also wrote, the typical change in cAMP level is a decline after CB1 activation, so please correct the figure accordingly. The presence of CB1 in microglia is questionable. Here the authors could somehow indicate high or low expression at the cell types. Indeed, by all cell -types the same four receptors are always written only with different colours. In this form, it is not very informative.
Response 6: Indeed, these are important considerations. As for the cAMP levels, the Reviewer’s view is likely neuronal and homo/monomeric on the CB1R. However, astrocytic CB1Rs are known to couple to Gq and increase astrocytic [Ca2+]i levels (e.g. Ferré et al., Trends Pharmacol Sci. 44(8):495-506. https://doi.org/10.1016/j.tips.2023.05.003.). To find the best solution for the figure, we showed the canonical downward arrow and in parenthesis, another, upward arrow.
As for receptor levels, we now take advantage of the + sign, and used them to represent the expected densities. Of course, one needs to take into account what the Reviewer above mentioned, i.e. the huge differences in CB1Rexpression in different neurons across brain areas. E.g. there is little CB1R expression in amygdalar GABAergic interneurons, while there is immense CB1R expression in hippocampal and cortical CCK+ interneurons. Also, the indication of expression/density levels are valid until neuroinflammation, when (micro)glial cells become reactive and their cannabinoid receptor expression can go up strongly. These limitations to the understanding of the figure have now been cautioned in the legend.
Comment 7: Line 151 Add that CBD activates GPR55.
Response 7: We have now expanded and better detailed the pharmacological actions of CBD at the four receptors in the legend to Figure 1, including GPR55. (Pag.6 lines 201-202)
Comment 8: Line 227 It seems to be trivial but could be mentioned that from the appearance of CB1 expression the developing embryonic brain is potentially sensitive to THC.
Response 8: This section was essentially focused on the role of endocannabinoid signaling in developing brain. Nevertheless, we added such mention suggested by the Reviewer, particularly important for readers from other fields of study (page 7, lines 298-299).
Comment 9: Lines 610-612 – It is worth to add that 2-AG is the major ligand of CB1, with 2-3 orders of magnitude higher concentration.
Response 9: This is indeed a good point. We added the suggested statement to section 1.2 where we felt to better belong. We also offer an interesting counterplay: anandamide’s basal level is indeed 2-3 orders of magnitude lower in brain homogenates than that of 2-AG (Buczynski and Parsons, 2010, Br J Pharmacol, DOI: 10.1111/j.1476-5381.2010.00787.x; and see also our paper: Köfalvi et al., 2015, Neuropharmacol, doi: 10.1016/j.neuropharm.2016.03.015.), but the difference between the “endocannabinoid signalling-competent” pools of 2-AG and anandamide are less than one order of magnitude.
We added the following sentence in page 2 lines 80-82: “It is worth noting that 2-AG is considered the principal eCB agonist in the brain, with signaling-competent levels that exceed those of anandamide by at least an order of magnitude [14,21]”
Minor points
Comment 10: Line 4 What indicates the number 3 after the author names?
Response 10: It was a mistake. It was corrected. We thank the Reviewer for noticing it.
Comment 11: Line 38-40 Delete “drug-naïve” – otherwise it is easy to misunderstand the sentence
Response 11: We feel necessary to keep “drug-naïve” because in subjects that have already been exposed to Δ9-THC, some of its effect can be the opposite, largely due to tolerance, neuroadaptations in the brain, and differences in the pharmacokinetics of the compound in chronic users. Here are some key distinctions. Drug-naïve subjects (mammals and humans) show strong responses across all four tests, with pronounced catalepsy, hypothermia, analgesia, and reduced locomotion. In contrast, regular consumers show diminished catalepsy, hyperthermia (!), reduced analgesia, and paradoxically increased locomotor activity.
Comment 12: Line 72 Delete “homodimer” – in their more usual monomeric form they also couple predominantly with Gi proteins.
Response 12: We have done it.
Comment 13: Line 77 Add “depending on cell type and area” because activation of CB1 does not lead activation of all these signaling pathways generally.
Response 13: We have done it.
Comment 14: Lines 199-202 Delete “earliest stage of nervous system development” it is written twice in the same sentence.
Response 14: We have corrected it.
Comment 15: Lines 524-529. -I think these sentences are misplaced in this chapter, they should appear at chapter 3.1 and 3.2
Response 15: We fully agree. It was used as an indication of a role of CB1R on synaptic pruning in PFC, but we now moved them to chapter 3 and discussed these studies in more detail. We thank the Reviewer for the suggestion.
Comment 16: Figure 2 – Again an excellent illustration. I suggest the change the colour of the oligodendrocyte, it is practically identical in the submitted version as the background.
Response 16: We modified it accordingly. Indeed, it is now more visible. We thank the Reviewer for the suggestion.
Comment 17: Lines 823-824 – Please change wording, because if the negative effect still remains (as written in line 823) than the study of Aguiar did not really allay it.
Response 17: We have corrected it.
Comment 18: Table 2 by FAAH D9 appears twice
Response 18: We have corrected it. We very much thank the Reviewer for noticing it.
Reviewer 2 Report
Comments and Suggestions for Authors
This is a long and well-written review that details the function and involvement of endocannabinoids and the endocannabinoid system in neural and brain development. The authors provide an in-depth look at the early to late stages of development at different organizational levels.
In addition, they point to the need for more longitudinal studies across the lifespan to fully understand the impact of the endocannabinoid system on brain development and function.
The table provides a nice summary of the action of key players of the endocannabinoid system.
Sometimes, the sentences are written awkwardly with an unusual grammar.
Section 2, page 6: some language editing will be helpful. For example: ‘where it was observed a more intense immunoreactivity for CB1R’. This statement does not seem right in terms of its grammar.
Section 2, page 7: ‘For instance, it has been elegantly shown the existence of a precise and concerted cellular and subcellular expression’ can be changed to ‘For instance, the existence of a precise and concerted cellular and subcellular expression has been elegantly shown’
Section 2, page 7: change ‘Interestingly, in opposition to the observed for CB1R, its expression decreases with differentiation [112].’ to ‘Interestingly, in contrast to the finding observed for CB1R, its expression decreases with differentiation [112].’
Section 2.1, line 289: change ‘In fact, the evidence pointing for an absence’ to ‘In fact, the evidence pointing to an absence’
Section 2.1, line 350: change ‘since WIN55,212,2 exposure from E5 in rats induced and increase in the number of GABA cells’ to ‘since WIN55,212,2 exposure from E5 in rats induced an increase in the number of GABA cells’
Section 2.2, line 408: change ‘This achieved by a chemorepulsion action of CB1R at the actin-rich growth cone, including motile filopodial extensions, driving growth cone steering’ to ‘This is achieved by a chemorepulsion action of CB1R at the actin-rich growth cone, including motile filopodial extensions, driving growth cone steering’
Section 2.2, line 473: change ‘this was show to induce’ to ‘this was shown to induce’
Line 683: change ‘This includes increased in FAAH’ to ‘This includes an increase in FAAH’
Line 723: change 'CB1Rs in the this brain area’ to ‘CB1Rs in this brain area’
Author Response
Comment 1: This is a long and well-written review that details the function and involvement of endocannabinoids and the endocannabinoid system in neural and brain development. The authors provide an in-depth look at the early to late stages of development at different organizational levels.
In addition, they point to the need for more longitudinal studies across the lifespan to fully understand the impact of the endocannabinoid system on brain development and function.
The table provides a nice summary of the action of key players of the endocannabinoid system.
Response 1: We very much thank the Reviewer for the time spent to comprehensively review our manuscript, for the positive comments and for the suggestions/corrections pointed out, which contributed to improve the quality of the manuscript. We have addressed them and we also have proofread the manuscript. The modifications are highlighted using track changes.
Comments 2: Sometimes, the sentences are written awkwardly with an unusual grammar.
Section 2, page 6: some language editing will be helpful. For example: ‘where it was observed a more intense immunoreactivity for CB1R’. This statement does not seem right in terms of its grammar.
Section 2, page 7: ‘For instance, it has been elegantly shown the existence of a precise and concerted cellular and subcellular expression’ can be changed to ‘For instance, the existence of a precise and concerted cellular and subcellular expression has been elegantly shown’
Section 2, page 7: change ‘Interestingly, in opposition to the observed for CB1R, its expression decreases with differentiation [112].’ to ‘Interestingly, in contrast to the finding observed for CB1R, its expression decreases with differentiation [112].’
Section 2.1, line 289: change ‘In fact, the evidence pointing for an absence’ to ‘In fact, the evidence pointing to an absence’
Section 2.1, line 350: change ‘since WIN55,212,2 exposure from E5 in rats induced and increase in the number of GABA cells’ to ‘since WIN55,212,2 exposure from E5 in rats induced an increase in the number of GABA cells’
Section 2.2, line 408: change ‘This achieved by a chemorepulsion action of CB1R at the actin-rich growth cone, including motile filopodial extensions, driving growth cone steering’ to ‘This is achieved by a chemorepulsion action of CB1R at the actin-rich growth cone, including motile filopodial extensions, driving growth cone steering’
Section 2.2, line 473: change ‘this was show to induce’ to ‘this was shown to induce’
Line 683: change ‘This includes increased in FAAH’ to ‘This includes an increase in FAAH’
Line 723: change 'CB1Rs in the this brain area’ to ‘CB1Rs in this brain area’
Response 2: We made all the corrections highlighted by the Reviewer. We thank very much the Reviewer for its contribution to improve the quality of the manuscript.
Reviewer 3 Report
Comments and Suggestions for Authors
This is an interesting and well-written review article on the neuromodulatory and pro-homeostatic role exerted by the endocannabinoid system during all critical periods of brain development, including from the embryogenesis up to the adolescence period. This is a topic already reviewed in the last years, but in articles concentrated on specific processes and/or specific developmental periods, not covering the whole period of brain development as the present review. In my opinion, it will be a useful text for those working in the field but also for those simply interested in this topic.
Specific comments:
1. Lines 49-50: the sentence is unclear as subcellular compartments are all intracellular.
2. Line 58: Epidiolex, although formulated with pure (90%) cannabidiol isolated from Cannabis sativa, is not usually considered “medical cannabis”, a term that refers more to formulations derived from cannabis extracts constituted by different plant-derived cannabinoids.
3. Line 127: FAAH enzyme is also able to degrade 2-AG.
4. References: some are duplicated and others with relevant data are missing. I would recommend to re-check the literature published on this topic.
Author Response
Comment 1: This is an interesting and well-written review article on the neuromodulatory and pro-homeostatic role exerted by the endocannabinoid system during all critical periods of brain development, including from the embryogenesis up to the adolescence period. This is a topic already reviewed in the last years, but in articles concentrated on specific processes and/or specific developmental periods, not covering the whole period of brain development as the present review. In my opinion, it will be a useful text for those working in the field but also for those simply interested in this topic.
Response 1: We very much thank the Reviewer for the time taken to review our manuscript and for sharing our idea of the relevance of a broader review summarizing the current knowledge on the role of endocannabinoid signaling in brain development. We also thank for the comments that contributed to improve the manuscript.
Specific comments:
Comment 2: Lines 49-50: the sentence is unclear as subcellular compartments are all intracellular.
Response 2: We agree. We have changed it to: “Both can activate CB1R and CB2R, which are located in various cell types and at different subcellular regions in neurons, at the cell surface and intracellularly”
Comment 3: Line 58: Epidiolex, although formulated with pure (90%) cannabidiol isolated from Cannabis sativa, is not usually considered “medical cannabis”, a term that refers more to formulations derived from cannabis extracts constituted by different plant-derived cannabinoids.
Response 3: This is true. We have altered the sentence as follows: “…important targets for medical cannabis formulations and Epidiolex, an antiepileptic medication based on CBD” (page 3 line 91). We thank the Reviewer for the correction
Comment 4: Line 127: FAAH enzyme is also able to degrade 2-AG.
Response 4: Indeed, there was a paper from Ken Mackie’s lab (Straiker et al. 2011 Br. J. Pharmacol https://doi.org/10.1111/j.1476-5381.2011.01486.x) showing that when they overexpressed FAAH in hippocampal culture and gauged 2-AG degradation indirectly, via DSE decay, they could observe that overexpression of FAAH impaired DSE. However, in a later review about 2-AG metabolism, the same authors claimed no significant effect of FAAH on 2-AG degradation under normal conditions in situ (Murataeva et al. Parsing the players: 2-arachidonoylglycerol synthesis and degradation in the CNS. Br J Pharmacol. 2014 171(6):1379-91. https://doi.org/10.1111/bph.12411. ).
Comment 5: References: some are duplicated and others with relevant data are missing. I would recommend to re-check the literature published on this topic.
Response 5: We very much thank the Reviewer for noticing the duplications, which we corrected now. We also added and discussed additional studies also in the reply to other Reviewer’s comments.